# Defective folate metabolism causes germline epigenetic instability and distinguishes *Hira* as a phenotype inheritance biomarker

Georgina E. T. Blake[1,2,4], Xiaohui Zhao ⬡ [1,2], Hong wa Yung[1,2], Graham J. Burton ⬡ [1,2], Anne C. Ferguson-Smith[2,3], Russell S. Hamilton ⬡ [2,3] & Erica D. Watson ⬡ [1,2✉]

The mechanism behind transgenerational epigenetic inheritance is unclear, particularly through the maternal grandparental line. We previously showed that disruption of folate metabolism in mice by the *Mtrr* hypomorphic mutation results in transgenerational epigenetic inheritance of congenital malformations. Either maternal grandparent can initiate this phenomenon, which persists for at least four wildtype generations. Here, we use genome-wide approaches to reveal genetic stability in the *Mtrr* model and genome-wide differential DNA methylation in the germline of *Mtrr* mutant maternal grandfathers. We observe that, while epigenetic reprogramming occurs, wildtype grandprogeny and great grandprogeny exhibit transcriptional changes that correlate with germline methylation defects. One region encompasses the *Hira* gene, which is misexpressed in embryos for at least three wildtype generations in a manner that distinguishes *Hira* transcript expression as a biomarker of maternal phenotypic inheritance.

[1] Department of Physiology, Development and Neuroscience, University of Cambridge, Cambridge, UK. [2] Centre for Trophoblast Research, University of Cambridge, Cambridge, UK. [3] Department of Genetics, University of Cambridge, Cambridge, UK. [4] Present address: College of Medicine and Health, University of Exeter Medical School, Exeter, UK. ✉email: edw23@cam.ac.uk

Environmental stressors can impact an individual's health and that of their progeny[1–5]. The phenotypic risk that persists for several generations in the absence of the stressor is termed transgenerational epigenetic inheritance (TEI)[6]. Although the mechanism is unclear, this non-conventional inheritance likely occurs independent of DNA base-sequence mutations and involves the inheritance of an epigenetic factor(s) via the germline[6,7]. Candidate factors in mammals include DNA methylation, histone modifications, and/or non-coding RNA[1,2,4,8–10]. How an epigenetic message resists reprogramming and is transmitted between one or even multiple generations to cause disease remains elusive. Few mammalian models of TEI exist and most focus on paternal inheritance[1,2,8–11]. We previously reported the $Mtrr^{gt}$ mouse line, a rare model of maternal grandparental TEI in which congenital malformations are transgenerationally inherited for at least four wildtype generations[3] (see below, Supplementary Fig. 1a, d–e). While the germline is implicated, the epigenetic mechanism remains unclear.

MTRR (methionine synthase reductase) is a key enzyme required for one-carbon metabolism (i.e., folate and methionine metabolism; Supplementary Fig. 2)[12–14]. Folate is a well-known vitamin important for neural tube closure, yet its function in development is complex and poorly understood. One-carbon metabolism is required for thymidine synthesis[15] and cellular methylation. Indeed, it transmits methyl groups for the methylation of homocysteine by methionine synthase (MTR) to form methionine and tetrahydrofolate[16]. Methionine acts as a precursor for $S$-adenosylmethionine (SAM), which serves as the sole methyl-donor for substrates involved in epigenetic regulation (e.g., DNA, RNA, histones)[17–19]. MTRR activates MTR through the reductive methylation of its vitamin $B_{12}$ cofactor[14] (Supplementary Fig. 2). Consequently, the progression of one-carbon metabolism requires MTRR to maintain genetic and epigenetic stability.

The hypomorphic $Mtrr^{gt}$ mutation reduces wildtype $Mtrr$ transcript expression in mice[3,12], which is sufficient to diminish MTR activity by ~60% (ref. [12]). Consequently, the progression of one-carbon metabolism is inhibited by the $Mtrr^{gt}$ mutation. Similar to mutations in the human $MTRR$ gene[13,20–22] or dietary folate deficiency in humans[23], $Mtrr^{gt/gt}$ mice display hyperhomocysteinemia[3,12] and macrocytic anaemia[24] in adulthood, as well as altered DNA methylation patterns associated with gene misexpression[3] and a broad range of incompletely penetrant developmental phenotypes at midgestation (e.g., growth defects and/or congenital malformations including heart, placenta, and neural tube closure defects)[3]. Therefore, $Mtrr^{gt}$ mice are suitable for studying defective folate metabolism.

Crucially, the $Mtrr^{gt}$ mouse line is a model of TEI that occurs via the maternal grandparental lineage[3]. Through highly controlled genetic pedigrees (Supplementary Fig. 1a, d–e), we demonstrated that an $Mtrr^{+/gt}$ heterozygous male or female mouse (i.e., the F0 generation) can initiate TEI of developmental phenotypes at embryonic day (E) 10.5 in the wildtype ($Mtrr^{+/+}$) descendants until the F4 generation[3]. This phenomenon occurs when all mice are wildtype for the $Mtrr$ gene except for the initiating F0 $Mtrr^{+/gt}$ individual (Supplementary Fig. 1d–e). The phenotypes are similar in kind to those observed in $Mtrr^{gt/gt}$ conceptuses (see above), though present at slightly lower frequencies[3]. Also, the phenotypes associate with locus-specific changes in DNA methylation that are linked to gene misexpression[3] giving functional relevance to this epigenetic disruption.

Regardless of whether an F0 $Mtrr^{+/gt}$ male or female initiates TEI, the spectrum and frequency of developmental phenotypes in the F2-F4 wildtype generations are largely comparable between pedigrees when inherited via their daughters[3]. The exception is

the F1 generation where phenotypic risk at E10.5 occurs only when F1 individuals are derived from F0 $Mtrr^{+/gt}$ females (Supplementary Fig. 1d–e)[3]. That is, the F1 progeny of F0 $Mtrr^{+/gt}$ males do not exhibit phenotypes at E10.5, reinforcing that these phenotypes are not transmitted through the male lineage (Supplementary Fig. 1d–e)[3]. However, F1 wildtype mice derived from F0 $Mtrr^{+/gt}$ males display indicators of direct epigenetic inheritance including locus-specific epigenetic dysregulation in placentas at E10.5 associated with gene misexpression in the absence of gross phenotype[3], a hematopoietic phenotype later in life[24], and the ability of F1 wildtype females to perpetuate epigenetically inherited phenotypes to their offspring in a manner similar to those derived from an F0 $Mtrr^{+/gt}$ female (Supplementary fig. 1d–e)[3].

Since TEI in the $Mtrr^{gt}$ model implicates the maternal lineage, we previously performed a blastocyst transfer experiment to show that phenotype inheritance occurred via the germline and was independent of the uterine environment. More specifically, F2 wildtype blastocysts derived from an F0 $Mtrr^{+/gt}$ maternal grandparent and F1 wildtype mother were transferred into the uteri of control females (Supplementary Fig. 1f)[3]. The risk for congenital malformations persisted after blastocyst transfer, indicating that phenotypic inheritance was not attributed to the uterine environment of the original mother but instead to the inheritance of an unknown epigenetic factor via the germline[3]. Therefore, we hypothesise that wildtype gametes from either $Mtrr^{+/gt}$ maternal grandparent can initiate phenotypic inheritance via the F1 wildtype daughters for several generations[3]. However, the mechanism(s) of TEI remains unclear in the $Mtrr^{gt}$ model including the identity of the inherited epigenetic factor(s) and whether there are different epigenetic cues in the F0 male and female germlines that can initiate similar effects in their grandprogeny.

Here, we investigate the potential mechanism(s) of TEI in the $Mtrr^{gt}$ model using a genome-wide approach. First, we demonstrate that $Mtrr^{gt/gt}$ mice are genetically stable and hence reassert focus on an epigenetic mechanism. Second, we show that germline DNA methylation is altered in F0 $Mtrr^{+/gt}$ males. F0 sperm were chosen for analysis because: (i) F0 $Mtrr^{+/gt}$ males initiate TEI of phenotypes in a similar manner to F0 $Mtrr^{+/gt}$ females, (ii) sperm are more experimental tractable than oocytes, and (iii) when assessing heritable effects, the uterine environment does not need to be controlled for in F0 $Mtrr^{+/gt}$ males. Even though differentially methylated regions (DMRs) in sperm of F0 $Mtrr^{+/gt}$ males are reprogrammed in somatic tissue of wildtype F1 and F2 progeny, our data shows evidence of transcriptional changes associated with germline epigenetic disruption that persists at least until the F3 generation. This proposed transcriptional memory of sperm DMRs includes misexpression of $Hira$, a gene important for chromatin stability[25,26] and production of rRNA[27], which we propose as a biomarker and potential mediator of maternal phenotypic inheritance in the $Mtrr^{gt}$ model.

## Results

**Genetic stability in $Mtrr^{gt}$ mice.** As one-carbon metabolism is directly linked to DNA synthesis[15], we first addressed whether the $Mtrr^{gt}$ allele influences genetic stability. Whole-genome sequencing (WGS) was performed on phenotypically normal C57Bl/6 J control embryos ($N = 2$) and $Mtrr^{gt/gt}$ embryos with congenital malformations ($N = 6$) derived from $Mtrr^{gt/gt}$ intercrosses (Supplementary Fig. 1a, c). DNA libraries were sequenced separately resulting in ~30× coverage per embryo (~3.5 × 10^8 paired-end reads/genome). The sequenced genomes were compared to the C57Bl/6 J reference genome to identify structural variants (SVs) and single-nucleotide polymorphisms (SNPs).

The $Mtrr^{gt}$ mutation was generated by a gene-trap (gt) insertion into intron nine of the $Mtrr$ gene (Chr13) in the 129P2Ola/Hsd mouse strain before eight generations of back-crossing into the C57Bl/6 J strain[3]. As a result, the majority of variants identified in $Mtrr^{gt/gt}$ embryos were located on Chr13 in the genomic region surrounding the $Mtrr$ locus (Supplementary Fig. 3a–b). These variants included the gene-trap and several SNPs that showed sequence similarity to the 129P2Ola/Hsd genome and likely persisted due to $Mtrr^{gt}$ genotype selection and regional crossover frequency. Variant identification in this region acted as an internal positive control of our bioinformatic method, demonstrating that it is capable of distinguishing genetic differences between experimental groups. Using these SNPs, we defined a 20 Mb region of 129P2Ola/Hsd sequence surrounding the $Mtrr^{gt}$ allele (Fig. 1a). When this region was bioinformatically masked, C57Bl/6 J and $Mtrr^{gt/gt}$ embryos contained a similar mean (±sd) frequency of SNPs (C57Bl/6 J: $4,871 \pm 791$ SNPs/embryo; $Mtrr^{gt/gt}$: $5138 \pm 398$ SNPs/embryo; $p = 0.781$) and SVs (C57Bl/6 J: 342 SVs/embryo; $Mtrr^{gt/gt}$: 301 SVs/embryo; $p = 0.6886$; Fig. 1b, c) implying that the de novo mutation rate was unchanged by the $Mtrr^{gt/gt}$ mutation. These values were in line with expected de novo mutation rates[28]. Only 25 (21 SNPs and 4 SVs) variants were present in all six $Mtrr^{gt/gt}$ embryos and absent in C57Bl/6 J embryos (Supplementary Fig. 3e, f, Supplementary Tables 1–2). When all SNPs and SVs were considered, the majority represented non-coding variants or were located in non-coding regions (Supplementary Fig. 3c, d). Moreover, genetic variation within the masked region had the minimal functional effect (beyond the gene-trap insertion) since no variant over-lapped with a known enhancer, and expression of individual genes was similar among C57Bl/6 J, 129P2Ola/Hsd and $Mtrr^{gt/gt}$ mice (Fig. 1a, d). Genomic stability was further supported by the preserved repression of transposable elements[29,30] in $Mtrr^{gt/gt}$ tissue (Fig. 1e) despite global DNA hypomethylation caused by the $Mtrr^{gt/gt}$ mutation[3]. Overall, these data support genetic integrity within the $Mtrr^{gt}$ model, and that phenotypic inheritance was unlikely caused by an increased frequency of de novo mutation. Therefore, the focus shifted to an epigenetic mechanism.

**Germline DNA methylation is altered in the $Mtrr^{gt}$ model.** MTRR protein has a direct role in the transmission of one-carbon methyl groups for DNA methylation[3,12,14]. Therefore, germline DNA methylation was considered as a potential mediator of phenotype inheritance. As an $Mtrr^{+/gt}$ female or male can initiate TEI (Supplementary Fig. 1d–e)[3] and due to the experimental tractability of male gametes, we focussed our analysis on sperm. Spermatogenesis and male fertility are normal in $Mtrr^{+/+}$, $Mtrr^{+/gt}$ and $Mtrr^{gt/gt}$ males[31]. Mature spermatozoa were collected from caudal epididymides of C57Bl/6 J, $Mtrr^{+/+}$, $Mtrr^{+/gt}$ and $Mtrr^{gt/gt}$ mice (Supplementary Fig. 1a–c) and sperm purity was confirmed by assessing imprinted regions of known methylation status via bisulfite pyrosequencing (Supplementary Fig. 4a). Global 5-methylcytosine (5mC) and 5-hydroxymethylcytosine (5hmC) levels were consistent across all $Mtrr$ genotypes relative to C57Bl/6 J control as determined by mass spectrometry (Fig. 2a).

To analyse the genome-wide distribution of sperm DNA methylation, methylated DNA Immunoprecipitation (IP) followed by sequencing (MeDIP-seq) was performed. This approach allowed the unbiased detection of locus-specific changes in DNA methylation by identifying clusters of differentially methylated cytosines, thus reducing the potential impact of single-nucleotide variants[4,32]. MeDIP libraries of sperm DNA were prepared using eight males each from C57Bl/6 J, $Mtrr^{+/+}$, $Mtrr^{+/gt}$ and $Mtrr^{gt/gt}$

genotypes (Supplementary Fig. 4b). Sequencing generated 179 million paired-end mappable reads on average per group (C57Bl/6 J: 164 million reads; $Mtrr^{+/+}$: 172 million reads; $Mtrr^{+/gt}$: 203 million reads; $Mtrr^{gt/gt}$: 179 million reads). Using MEDIPS package[33], each $Mtrr$ genotype was independently compared to C57Bl/6 J controls. Loci of >500 bp with a methylation change of >1.5-fold and $p < 0.01$ were defined as DMRs. The number of sperm DMRs identified increased with the severity of $Mtrr$ genotype: 91 DMRs in $Mtrr^{+/+}$ males, 203 DMRs in $Mtrr^{+/gt}$ males and 599 DMRs in $Mtrr^{gt/gt}$ males (Fig. 2b, c). The presence of DMRs in sperm from $Mtrr^{+/+}$ males indicated a parental effect of the $Mtrr^{gt}$ allele on offspring germline methylome since $Mtrr^{+/+}$ males derive from $Mtrr^{+/gt}$ intercrosses (Supplementary Fig. 1b). Hypo- and hypermethylated regions were identified in each $Mtrr$ genotype when compared with C57Bl/6 J controls (Fig. 2c), consistent with earlier findings in placentas[3]. These data suggested that the $Mtrr^{gt}$ allele was sufficient to dysregulate sperm DNA methylation.

To ensure the robustness and reliability of the MeDIP-seq data, we randomly selected hyper- and hypomethylated DMRs to validate using bisulfite pyrosequencing. Sperm DNA from C57Bl/6 J, $Mtrr^{+/+}$, $Mtrr^{+/gt}$ and $Mtrr^{gt/gt}$ males was assessed ($N = 8$ males/group: four sperm samples from MeDIP-seq experiment plus four independent samples). DMRs were validated in the $Mtrr$ genotype in which they were identified (Figs. 2d, 3, Supplementary Fig. 5). The overall validation rate was 94.1% in hypomethylated DMRs and 58.3% in hypermethylated DMRs (Supplementary Table 3) and indicated a high degree of corroboration between techniques. The majority of DMRs that did not validate showed extensive methylation (>80% CpG methylation) in C57Bl/6 J sperm and were identified as hypermethylated in the MeDIP-seq experiment (Supplementary Fig. 5). This might reflect some false positives in line with another study[4].

For most DMRs assessed, methylation change was consistent across all CpG sites and the absolute change in CpG methylation ranged from 10 to 80% of control levels (Figs. 2d, 3, Supplementary Fig. 5). Within each genotypic group, a high degree of inter-individual consistency of methylation change was also observed. Therefore, we conclude that the $Mtrr^{gt}$ mutation, or parental exposure to it as in $Mtrr^{+/+}$ males, is sufficient to lead to distinct DNA methylation changes in sperm.

**Most DMRs associate with metabolic dysregulation, not genetic effects.** A proportion of the DMRs was located within the region around the gene-trap insertion site in $Mtrr^{+/gt}$ and $Mtrr^{gt/gt}$ males (Fig. 1a, Supplementary Fig. 6b, c), consistent with $Mtrr^{gt/gt}$ liver[34] and suggesting that the gene-trap or underlying 129P2Ola/Hsd sequence might epigenetically dysregulate the surrounding region. However, a comparison of the MeDIP-seq and WGS data sets revealed that genetic variation did not influence DMR calling to a great extent since only a small proportion (2.8–5.5%) of these DMRs contained one or more SNP. Eight DMRs overlapped with known enhancers (Supplementary Table 4), none of which associated with promoters[35] containing a genetic variant. Outside of the $Mtrr$ genomic region, 54 DMRs were common to $Mtrr^{+/+}$, $Mtrr^{+/gt}$ and $Mtrr^{gt/gt}$ males (Fig. 2b, c, Supplementary Table 5) and were primarily located in distinct chromosomal clusters (Supplementary Fig. 6a–d). These data implicate epigenetic hot-spots or underlying genetic effects. However, beyond a polymorphic duplication on Chr19 in the C57Bl/6 J strain[36] that accounted for a minor number of DMRs (2.5–15.8% of DMRs), no DMRs overlapped with an SV or were located <1 kb of an SV. Once potential genetic effects were accounted for, the majority of sperm DMRs in $Mtrr^{+/+}$ and $Mtrr^{+/gt}$ males (76/91 DMRs and 142/203 DMRs, respectively), and a proportion of sperm DMRs in

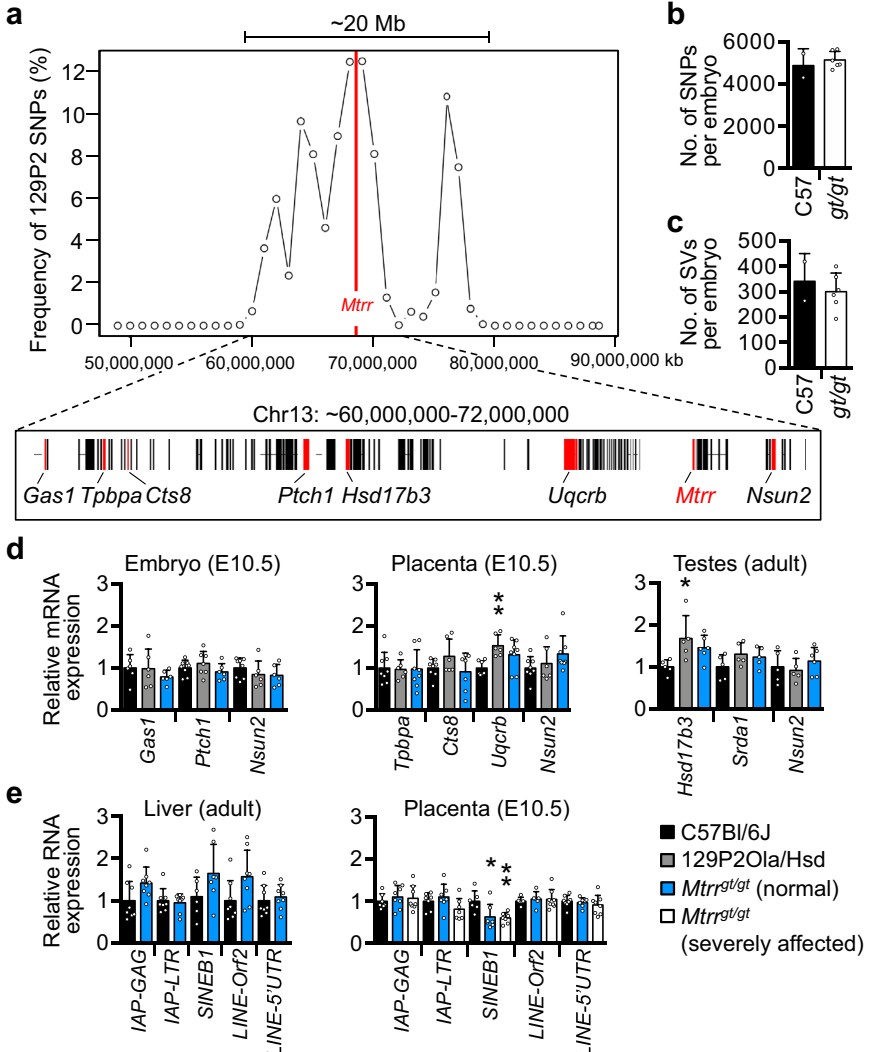

**Fig. 1 The _Mtrr^gt_ mouse line is genetically stable. a–c** Whole-genome sequencing (WGS) of normal C57Bl/6 J embryos ($N = 2$ embryos) and severely affected _Mtrr^gt/gt_ embryos ($N = 6$ embryos) at embryonic day (E) 10.5 to determine the frequency of genetic variants compared to the C57Bl/6 J reference genome. **a** The frequency of 129P2Ola/Hsd (129P2) single-nucleotide polymorphisms (SNPs) in the region surrounding the gene-trap insertion site in the _Mtrr_ gene (red line) on chromosome 13 (Chr13). The majority of genes within the 20 megabase (Mb) region surrounding the _Mtrr_ gene are shown below the graph. **b**, **c** The average number of **b** SNPs and **c** structural variants (SVs) per embryo (mean ± standard deviation [sd]) in C57Bl/6 J embryos (C57, black bars) and _Mtrr^gt/gt_ embryos (_gt/gt_, white bars). Two-tailed unpaired _t_ test with Welch's correction. The 20 Mb region shown in **a** was masked when calculating the average number of genetic variants in **b**, **c**. **d** Graphs showing RT-qPCR analysis of selected genes (highlighted red in **a**) in embryos (E10.5; $N = 6$–8 embryos/group), placentas (E10.5; $N = 6$–8 placentas/group) and/or adult testes ($N = 5$–6 males/group) from C57Bl/6 J (black bars), 129P2Ola/Hsd (grey bars) and phenotypically normal _Mtrr^gt/gt_ (blue bars) mice. One-way ANOVA with Tukey's multiple comparison test, $*p = 0.022$, $**p = 0.007$. **e** Graphs showing RNA expression of specific groups of transposable elements as determined by RT-qPCR in adult liver ($N = 6$–8 livers/group) and placentas (E10.5; $N = 6$–7 placentas/group) from C57Bl/6 J mice (black bars) and _Mtrr^gt/gt_ mice (blue bars: phenotypically normal; white bars: severely affected). Two-way ANOVA with Tukey's multiple comparison test, $*p = 0.011$, $**p = 0.005$. Data from RT-qPCR analyses in **d**, **e** are shown as mean ± sd and relative to C57Bl/6 J tissue levels (normalised to 1). Source data are provided as a Source Data file.

_Mtrr^gt/gt_ males (174/599 DMRs) were attributed to the long-term metabolic consequences of the _Mtrr^gt_ mutation.

**Sperm DMR genomic distribution and potential regulatory function.** DMR distribution was determined to explore the regional susceptibility of the sperm methylome to the effects of the _Mtrr^gt_ allele. First, the sperm 'background methylome' was established to resolve the expected genome-wide distribution of CpG methylation (see Methods). By comparing the regional distribution of sperm DMRs to the background methylome, we revealed that DMRs in all _Mtrr_ genotypes were not significantly enriched in repetitive regions (Fig. 2e). However, sperm DMRs in _Mtrr^+/+_ and _Mtrr^+/gt_ males were over-represented in introns and

exons, and under-represented in intergenic regions ($p < 0.0003$, Chi-squared test; Fig. 2f). This was not the case for _Mtrr^gt/gt_ males since DMRs were proportionally distributed among most genomic regions (Fig. 2f). Although the majority of sperm DMRs were located within CpG deserts, a proportion of DMRs from _Mtrr^+/gt_ and _Mtrr^gt/gt_ males were enriched in CpG islands ($p < 0.0014$, Chi-squared; Fig. 2g), which has implications for gene regulatory control. Finally, when considering only the subset of common DMRs shared by all _Mtrr_ genotypes, a similar genomic distribution to _Mtrr^+/+_ and _Mtrr^+/gt_ males was observed (Supplementary Fig. 6d, e).

During sperm maturation, histones are replaced by protamines[37]. However, ~1% of histone-containing nucleosomes

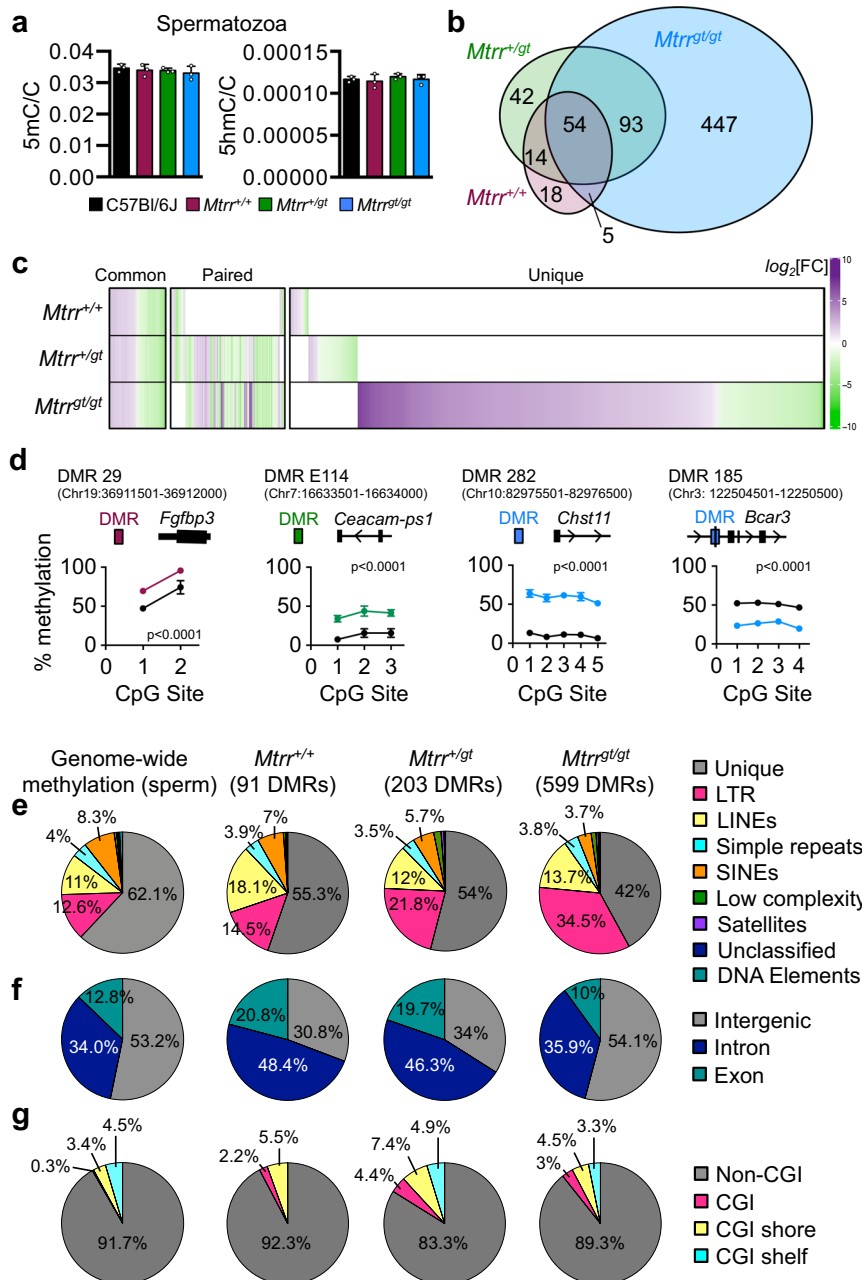

are retained[38] providing scope for epigenetic inheritance[39]. Nucleosome retention occurs primarily at promoters of developmentally regulated genes and gene-poor repeat regions, though regional distribution and frequency differ between reports[40,41]. To determine whether sperm DMRs in *Mtrr* males were enriched in known sites of nucleosome retention, we utilised the MNase-seq data set from C57Bl/6 J spermatozoa in Erkek et al.[41]. First, by randomly selecting 10,000 regions of 500 bp as a proxy for DMRs, we determined that the expected frequency of DMR overlap with sites of nucleosome retention was 1.94%. Crucially, we observed that 14.5–34.1% of DMRs identified in sperm from *Mtrr*[+/+], *Mtrr*[+/gt] and *Mtrr*[gt/gt] males were located in nucleosome retention regions (Table 1), indicating a significant enrichment ($p < 0.0001$, binomial test). Therefore, these DMRs represented candidate regions for epigenetic inheritance.

To better understand the normal epigenetic signatures within regions identified as sperm DMRs and to predict a potential gene

regulatory role, mean enrichment for histone modifications and/or chromatin accessibility in mouse spermatozoa[42], epiblast and extraembryonic ectoderm at E6.5[43] was determined using published ChIP-seq and ATAC-seq data sets. All DMRs, except those surrounding the *Mtrr* gene-trapped site, were analyzed ($N = 379$ DMRs from all *Mtrr* genotypes combined) alongside 379 randomly selected regions representing the 'baseline genome' (see Methods). Compared with the sperm baseline genome, the majority of our DMRs were likely to associate with a closed chromatin state due to collective enrichment for protamine 1 (PRM1) and repressive histone mark H3K9me3, but not active histone marks (e.g., H3K4me1, H3K27ac) or Tn5 transposase sensitive sites (THSS)[42] (Supplementary Fig. 7a, b). This finding reinforces heterogeneity of DMR association with retained nucleosomes[41] or protamines (Table 1). In contrast, the DMRs were more likely in an open chromatin conformation state in epiblast and extraembryonic ectoderm at E6.5 based on collective

**Fig. 2 Characterisation of differential DNA methylation in spermatozoa from *Mtrr^gt* mouse line. a** Global 5-methylcytosine (5mC) and 5-hydroxymethylcytosine (5hmC) in spermatozoa from C57Bl/6 J (black bars), wildtype (*Mtrr^+/+*; purple bars), *Mtrr^+/gt* (green bars) and *Mtrr^gt/gt* (blue bars) adult males (N = 9 males/genotype, analysed in three pools of three males/genotype) as assessed by mass spectrometry. Data are presented as the ratio of methylated cytosines per genomic cytosines (5mC/C or 5hmC/C; mean ± standard deviation [sd]). One-way ANOVA. **b–c** Methylated DNA immunoprecipitation followed by sequencing (MeDIP-seq) of spermatozoa DNA from *Mtrr^+/+*, *Mtrr^+/gt* and *Mtrr^gt/gt* males relative to C57Bl/6 J control males was performed to determine differentially methylated regions (DMRs). N = 8 males/group. **b** An intersectional analysis of DMRs in *Mtrr^+/+* (purple), *Mtrr^+/gt* (green), and *Mtrr^gt/gt* (blue) relative to C57Bl/6 J controls. **c** A heat map plotting *log₂*FoldChange (*log₂*[FC]) of CpG methylation in spermatozoa from *Mtrr^+/+*, *Mtrr^+/gt* and *Mtrr^gt/gt* males compared to C57Bl/6 J males (p < 0.05). DMRs that were common between all *Mtrr* genotypes, paired between two *Mtrr* genotypes, or unique to a single *Mtrr* genotype when compared to C57Bl/6 J controls are shown. Relative to C57Bl/6 J sperm: DMR hypermethylation shown in purple, DMR hypomethylation shown in green, normal DMR methylation shown in white. Darker colour intensity indicates greater differential methylation. **d** Examples of DMRs identified via MeDIP-seq and validated by bisulfite pyrosequencing in sperm from the male genotype in which the DMR was found. Data are shown as percentage methylation at each CpG site assessed (mean ± sd). C57Bl/6 J (black circles), *Mtrr^+/+* (purple circles), *Mtrr^+/gt* (green circles) or *Mtrr^gt/gt* (blue circles) males. N = 8 males/group including four males from the MeDIP-seq analysis and four independent males. DMR ID and chromosomal location are shown along with a schematic of each DMR in relation to the closest gene. Two-way ANOVA with Sidak's multiple comparisons tests performed on mean methylation per CpG site per genotype group, p < 0.0001 for all comparisons. See also Fig. 3 and Supplementary Fig. 5. **e–g** Relative distribution of methylated regions identified via MeDIP-seq in C57Bl/6 J sperm (background methylome) and sperm DMRs from *Mtrr^+/+*, *Mtrr^+/gt* and *Mtrr^gt/gt* males among **e** unique sequences and repetitive elements, **f** coding and non-coding regions and **g** CpG islands (CGIs), shores and shelves. Statistical analyses: **e, g** one-tailed Chi-squared test; **f** Two-way ANOVA with Dunnett's multiple comparison test. *LTR* long terminal repeats, *LINEs* long interspersed nuclear elements, *SINEs* small interspersed nuclear elements. Source data are provided as a Source Data file.

enrichment for THSS when compared with tissue-specific baseline genomes[43] (Supplementary Fig. 7c). Therefore, it is possible that the genomic regions identified as DMRs in *Mtrr* sperm have a regulatory role during development.

**Some DMRs were located in regions of reprogramming resistance.** DNA methylation is largely reprogrammed during pre-implantation development and in the developing germline[44,45] to 'reset' the methylome between each generation. Recently, several loci were identified as 'reprogramming resistant'[46–48] and thus, are implicated in epigenetic inheritance. Using published methylome data sets[46,47], we determined that 40.7–54.3% of sperm DMRs across all *Mtrr* genotypes fell within loci resistant to pre-implantation reprogramming (Table 1). Sixteen of these DMRs were common among *Mtrr^+/+*, *Mtrr^+/gt* and *Mtrr^gt/gt* males. Fewer DMRs correlated with regions resistant to germline reprogramming (2.2–3.8% of DMRs/*Mtrr* genotype; Table 1) or both pre-implantation and germline reprogramming (2.0–2.7% of DMRs/*Mtrr* genotype; Table 1). Only one DMR located in a region resistant to germline reprogramming was common to all *Mtrr* genotypes. Furthermore, several DMRs were characterised as regions of reprogramming resistance[46,47] and nucleosome retention[41] (Table 1). Overall, differential methylation of these key regions in sperm of *Mtrr^gt* males might have important implications for epigenetic inheritance.

**Sperm DMRs are reprogrammed in wildtype F1 and F2 generations.** TEI in the *Mtrr^gt* model occurs via the maternal grandparental lineage[3] (Supplementary Fig. 1d–e). To determine the heritability of germline DMRs, bisulfite pyrosequencing was used to validate 10 sperm DMRs from F0 *Mtrr^+/gt* males in the tissue of wildtype F1 and F2 progeny (i.e., the maternal grandfather pedigree). The breeding scheme was as follows: F0 *Mtrr^+/gt* males were mated with C57Bl/6 J control females. The resulting F1 progeny was either collected at E10.5 for analysis or allowed to litter out. In the latter case, adult F1 wildtype females were mated with C57Bl/6 J control males and the F2 wildtype progeny was collected at E10.5 for analysis (Supplementary fig. 1d). The advantage of assessing inheritance of DNA methylation in F0 sperm rather than in F0 oocytes was that potential confounding effects of the F0 uterine environment could be avoided. Candidate DMRs were hyper- or hypomethylated, and localised

to regions of reprogramming resistance and/or to intra- or intergenic regions (Supplementary Table 6). In general, all DMRs tested lost their differential methylation in wildtype F1 and F2 embryos and placentas at E10.5, and showed DNA methylation patterns similar to C57Bl/6 J tissue (Fig. 3, Supplementary Table 7). This result occurred even when wildtype F2 conceptuses displayed congenital malformations (Fig. 3). DMRs were also assessed in *Mtrr^gt/gt* conceptuses at E10.5 to determine whether these regions were capable of differential methylation outside of the germline. In a manner similar to sperm from *Mtrr^+/gt* males (Fig. 3), seven out of 10 DMRs were hypermethylated in *Mtrr^gt/gt* embryos and/or placentas compared to control conceptuses (Supplementary Fig. 8a–j). In this case, it was unclear whether DNA methylation in these regions resisted epigenetic reprogramming or was erased and abnormally re-established/maintained due to intrinsic *Mtrr^gt/gt* homozygosity. Overall, altered patterns of DNA methylation in sperm of *Mtrr^+/gt* males were not evident in somatic tissue of wildtype progeny and grand-progeny. This result was reminiscent of mouse models of parental exposure to environmental stressors (e.g., maternal under-nutrition[4], paternal folate deficiency[49], or paternal cigarette smoking[50]), which induced sperm DMRs associated with phenotypes in the direct offspring even though the DMRs were resolved in offspring somatic tissue. As a result, other epigenetic mechanisms (e.g., germline RNA content and/or histone modifications) with DNA methylation are implicated in phenotypic inheritance.

**Potential transcriptional memory of sperm DMRs.** A previous study in mice suggests that sperm DMRs can associate with perturbed transcription in offspring even when DNA methylation is re-established to normal levels[4]. To assess whether transcriptional memory associated with sperm DMRs occurred in the *Mtrr^+/gt* maternal grandfather pedigree, expression of six genes located in or near sperm DMRs from F0 *Mtrr^+/gt* males (Supplementary Table 6) was assessed in F1 and F2 wildtype individuals. Although all six genes displayed normal expression in F1 tissues (Fig. 4a–c), three of these genes including *Hira* (histone chaperone), *Cwc27* (spliceosome-associated protein) and *Tshz3* (transcription factor) were misexpressed in F2 wildtype embryos or adult livers compared to C57Bl/6 J controls (Fig. 4d–f). This result might reflect transcriptional memory of the associated

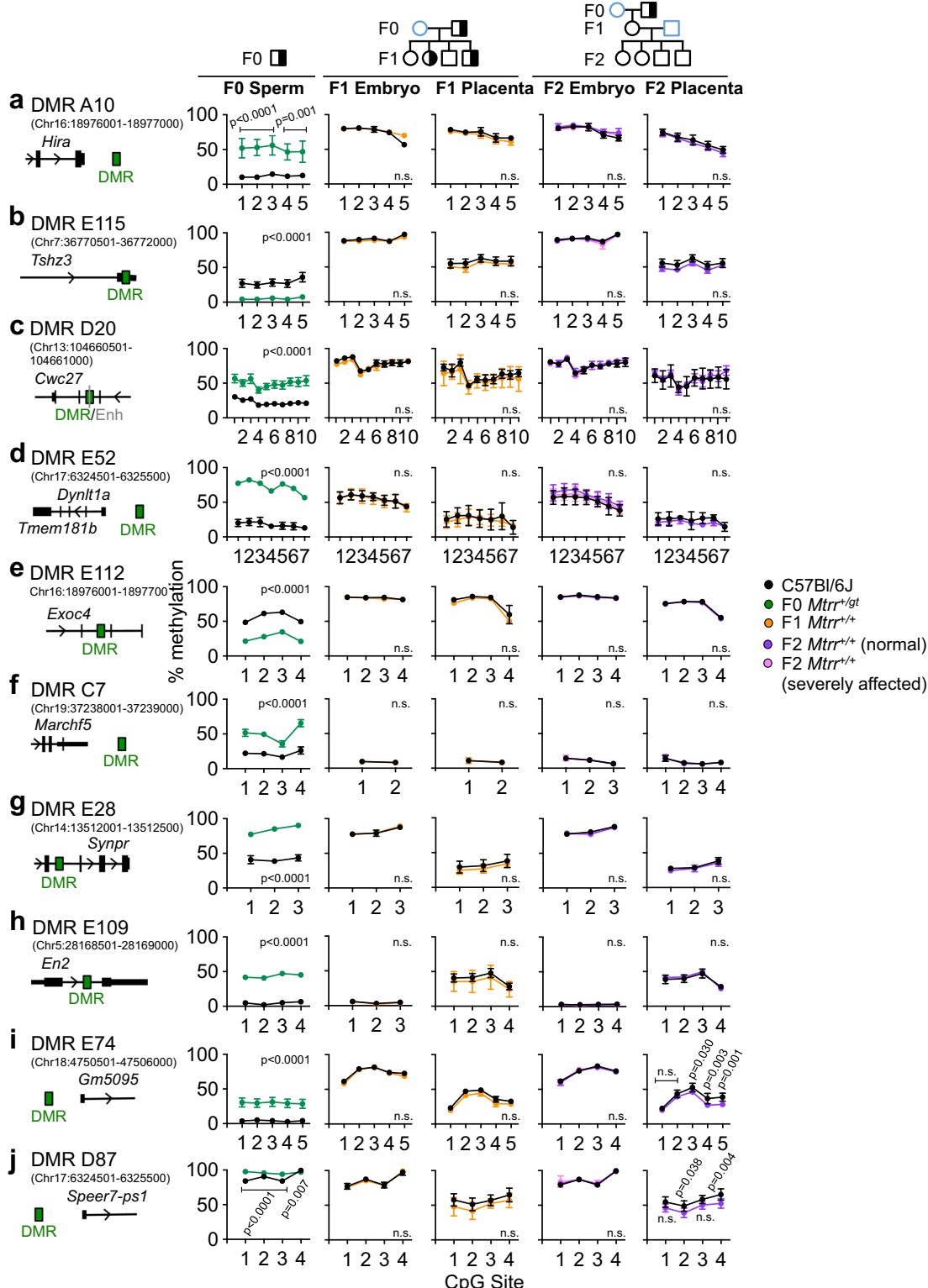

sperm DMR or wider epigenetic dysregulation in sperm of the F0 *Mtrr+/gt* males.

To further predict whether the *Hira*, *Cwc27* and *Tshz3* DMRs demarcate gene regulatory regions, their specific genetic and epigenetic characteristics beyond CpG methylation were considered. Genomically, the DMRs were located intragenically (*Cwc27* and *Tshz3* DMRs) or within 6 kb downstream of the gene (*Hira* DMR; Fig. 5a, Supplementary Figs. 9–10). Furthermore, *Cwc27* DMR overlapped with a known enhancer (Supplementary

Fig. 9) while *Hira* and *Tshz3* DMRs overlapped with CpG islands[51] (Fig. 5a, Supplementary Fig. 10). Next, we assessed the three DMRs for the enrichment of specific histone modifications during normal development using published ChIP-seq data sets[35] in wildtype embryonic stem cells (ESCs) and trophoblast stem cells (TSCs). Although histone marks were largely absent at all three DMRs in TSCs, enrichment for one or more methylated histone marks (e.g., H3K4me3 and/or H3K9me3) at this DMRs was apparent in ESCs (Fig. 5a, Supplementary Fig. 9–10).

**Fig. 3 Sperm DMRs are reprogrammed in somatic tissue of F1-F2 wildtype generations.** CpG methylation at specific sperm differentially methylated regions (DMRs) identified in F0 $Mtrr^{+/gt}$ males was assessed in the F1 and F2 wildtype embryos and placentas at E10.5. Pedigrees indicate a specific mating scheme (see also Supplementary Fig. 1a, 1d). F0, parental generation; F1 first filial generation; F2, second filial generation. **a–j** Schematic drawings of the sperm DMRs (green rectangles) assessed and the relationship to the closest gene and enhancer (Enh; grey line in **c**), if present. Graphs to the right in each panel show the average percentage of methylation at individual CpG sites for the corresponding DMR as determined by bisulfite pyrosequencing (mean ± standard deviation [sd]). In each graph, methylation was assessed in sperm from F0 $Mtrr^{+/gt}$ males (green circles), phenotypically normal F1 wildtype ($Mtrr^{+/+}$) embryos or placentas at E10.5 (orange circles), and phenotypically normal (purple circles) or severely affected (pink circles) F2 wildtype embryos or placentas at E10.5. C57Bl/6 J (black circles) for each tissue type are shown as controls. Specific N values for each DMR and experimental group are shown in Supplementary Table 7. In brief, the following N values were used (represented as a number of individuals/CpG site/ genotype): sperm: $N = 3$–8; F1 embryos: $N = 3$–5; F1 placentas: $N = 6$–8; F2 embryos: $N = 3$–5; F2 placentas: $N = 3$–8. Data are shown as mean ± sd for each CpG site. Two-way ANOVA, with Sidak's multiple comparisons test, performed on mean methylation per CpG site per genotype group. $p$ values indicated are for all CpG site comparisons within the DMR unless otherwise indicated. n.s., not significant. Pedigree legend: circle, female; square, male; blue outline, C57Bl/6 J control; black outline, $Mtrr^{gt}$ mouse line; white filled, $Mtrr^{+/+}$; half-white/half-black filled, $Mtrr^{+/gt}$. Source data are provided as a Source Data file.

**Table 1 Representation of sperm DMRs in candidate regions of epigenetic inheritance with in $Mtrr^{+/+}$, $Mtrr^{+/gt}$ and $Mtrr^{gt/gt}$ males.**

| Location of sperm DMRs | $Mtrr^{+/+}$ | $Mtrr^{+/gt}$ | $Mtrr^{gt/gt}$ |
|---|---|---|---|
| Genome-wide | 91 (100%)† | 203 (100%) | 599 (100%) |
| Regions of nucleosome retention[41] | 31 (34.1%) | 57 (28.1%) | 87 (14.5%) |
| Reprogramming resistant regions in pre-implantation embryo[47] | 37 (40.7%) | 96 (47.3%) | 325 (54.3%) |
| Reprogramming resistant regions in germline[46] | 2 (2.2%) | 5 (2.5%) | 23 (3.8%) |
| Reprogramming resistant regions in pre-implantation embryo[47] & germline[46] | 2 (2.2%) | 4 (2.0%) | 16 (2.7%) |
| Regions of nucleosome retention[41] & resistant to reprogramming in pre-implantation embryo[47] | 19 (20.9%) | 36 (17.7%) | 49 (8.9%) |
| Regions of nucleosome retention[41] & resistant to reprogramming in germline[46] | 1 (1.1%) | 2 (1.0%) | 4 (0.67%) |

†Number of DMRs identified compared to C57Bl/6 J controls followed by the percentage of the total number of DMRs in brackets.

Altogether, DMRs identified in sperm of $Mtrr^{+/gt}$ males highlight regions that might be important for transcriptional regulation in the early conceptus by other epigenetic mechanisms. Therefore, future analyses of broader epigenetic marks at these sites are required in the $Mtrr^{gt}$ mouse line.

**HIRA as a potential biomarker of maternal phenotypic inheritance.** The importance of the $Hira$ DMR in phenotypic inheritance was further considered based on its known resistance to germline reprogramming[46], and its potential function in gene regulation (Fig. 5a) and transcriptional memory (Fig. 4d). HIRA is a histone H3.3 chaperone, which lends itself well to a role in epigenetic inheritance given its broad functionality in transcriptional regulation[26], maintenance of chromatin structure in the developing oocyte[25] and the male pronucleus after fertilisation[27], and in rRNA transcription[27]. $Hira^{-/-}$ mice[52] and the $Mtrr^{gt}$ mouse line[3] display similar phenotypes including growth defects, congenital malformations, and embryonic lethality by E10.5. Furthermore, $Mtrr^{gt}$ genotypic severity correlated with the degree of hypermethylation in the $Hira$ DMR in sperm (Fig. 5b, c), suggesting that the $Hira$ DMR is responsive to alterations in folate metabolism.

The potential regulatory legacy of the $Hira$ DMR in sperm was further explored in the $Mtrr^{+/gt}$ maternal grandfather pedigree. The $Hira$ DMR, which was 6 kb downstream of the $Hira$ gene and overlaps with a CpG island and CTCF binding site in ESCs and TSCs (Fig. 5a), was substantially hypermethylated in sperm of F0 $Mtrr^{+/gt}$ males compared with controls ($39.0 \pm 4.1\%$ more methylated CpGs per CpG site assessed; Fig. 5b, c). As with the other sperm DMRs assessed (Fig. 3), the $Hira$ DMR was reprogrammed in F1–F3 wildtype embryos and placentas at E10.5 (Fig. 3a, Supplementary Fig. 11a). Although we originally assessed $Hira$ mRNA in the F1-F2 generations in the maternal grandfather pedigree (Fig. 4), further analysis revealed that $Hira$

isoforms (mRNA and long non-coding RNA (lncRNA 209); Fig. 5a) were differentially regulated at E10.5 (Fig. 6a–c). $Hira$ lncRNA function is unknown, though lncRNA-based mechanisms often control cell fates during development by influencing the nuclear organisation and transcriptional regulation[53]. Notably, this pattern of RNA expression was associated with generational patterns of phenotypic inheritance. For instance, we observed down-regulation of $Hira$ mRNA in F2–F3 wildtype embryos and not F1 wildtype embryos at E10.5 (Fig. 6a–c). Conversely, significant upregulation of $Hira$ lncRNA expression was apparent only in F1 wildtype embryos at E10.5 (Fig. 6a–c). This expression pattern was embryo-specific since the corresponding placentas showed normal $Hira$ transcript levels in each generation assessed compared to controls (Fig. 6a–b). Since phenotypes at E10.5 were apparent in F2 generation onwards, yet were absent in the F1 generation of the $Mtrr^{+/gt}$ maternal grandfather pedigree[3], embryo-specific $Hira$ RNA misexpression reflected the pattern of phenotypic inheritance.

To further investigate a potential link between $Hira$ expression and phenotypic inheritance, we analysed wildtype F1–F3 conceptuses at E10.5 derived from F0 $Mtrr^{+/gt}$ females (i.e., the maternal grandmother pedigree; Supplementary Fig. 1e). The breeding scheme was as follows: F0 $Mtrr^{+/gt}$ females were mated with C57Bl/6 J control males. The F1 progeny was collected at E10.5 for genotype and phenotype analysis or allowed to litter out. The resulting adult F1 wildtype females were mated with C57Bl/6 J males. The F2 wildtype progeny was similarly collected at E10.5 for analysis or adult F2 wildtype females were mated with C57Bl/6 J control males to generate F3 wildtype conceptuses for analysis at E10.5 (Supplementary Fig. 1e). In this pedigree, all generations, including the F1 generation, display a broad spectrum of developmental phenotypes[3]. Supporting our hypothesis, $Hira$ mRNA expression was downregulated in F1–F3 wildtype embryos derived from an F0 $Mtrr^{+/gt}$ female compared to C57Bl/6 J controls (Fig. 6d), thus correlating with maternal

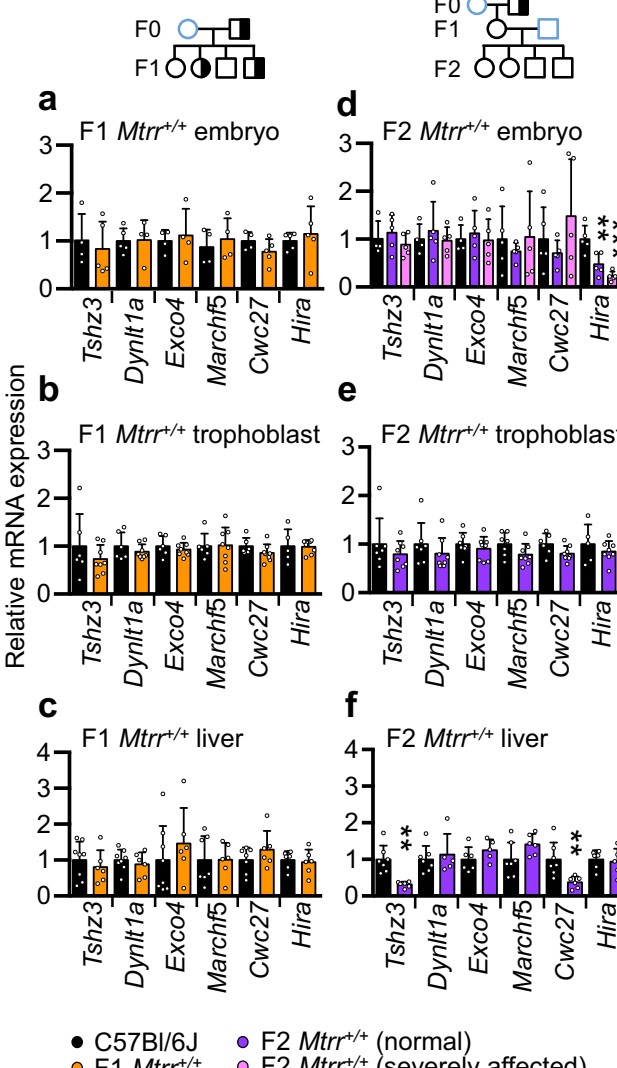

**Fig. 4 Transcriptional changes associated with some DMR-associated genes in F1 and F2 wildtype somatic tissue. a–f** RT-qPCR analysis of mRNA expression of genes located in close proximity to sperm DMRs in **a**, **d** embryos at E10.5, **b**, **e** placental trophoblast at E10.5, and **c**, **f** adult livers of wildtype (*Mtrr*+/+) F1 and F2 progeny. Tissue from phenotypically normal F1 wildtype individuals (orange bars) or F2 wildtype individuals (purple bars) or severely affected F2 wildtype individuals (pink bars) derived from an F0 *Mtrr*+/gt male was assessed. C57Bl/6 J control tissues (black bars) were assessed as controls. Pedigrees indicate specific mating schemes assessed (see also Supplementary Fig. 1a, d). Data are plotted as mean ± standard deviation, and are presented as relative expression to C57Bl/6 J levels (normalised to 1). Embryos: C57Bl/6 J, *N* = 4–5 embryos; F1 *Mtrr*+/+, *N* = 4–5 embryos; F2 *Mtrr*+/+ phenotypically normal or severely affected, *N* = 5 embryos. Trophoblast: C57Bl/6 J, *N* = 5–7 placentas; F1 or F2 *Mtrr*+/+, *N* = 8 placentas. Liver: C57Bl/6 J, *N* = 6–8 livers; F1 *Mtrr*+/+, *N* = 6 livers; F2 *Mtrr*+/+, *N* = 5–6 livers. Independent *t* tests or one-way ANOVA with Dunnett's multiple comparisons tests were performed. **d** *Hira* **\*\***$p$ = 0.0049, **\*\*\***$p$ = 0.0002; **f** *Tshz3* **\*\***$p$ = 0.0011; *Cwc27* **\*\***$p$ = 0.0095. Each gene was associated with the following sperm DMR shown in Fig. 3: *Tshz3*, DMR E115; *Dynlt1a*, DMR E52; *Exoc4*, DMR E112; *March5*, DMR C7; *Cwc27*, DMR D20; *Hira*, DMR A10. F0, parental generation; F1 first filial generation; F2, second filial generation. Pedigree legend: circle, female; square, male; blue outline, C57Bl/6 J control; black outline, *Mtrr*gt mouse line; white filled, *Mtrr*+/+; half-white/half-black filled, *Mtrr*+/gt. Source data are provided as a Source Data file.

phenotypic inheritance. As expected, *Hira* lncRNA transcripts were unchanged in F1 and F3 wildtype embryos (Fig. 6d). However, *Hira* lncRNA was downregulated in F2 wildtype embryos (Fig. 6d), which display the highest frequency of phenotypes among the three generations[3]. Regardless, these data suggested that altered *Hira* mRNA transcripts might be a potential biomarker of maternal phenotypic inheritance since dysregulation of *Hira* mRNA occurred only in wildtype embryos with high phenotypic risk as a result of their derivation from an oocyte with *Mtrr*gt ancestry rather than sperm (Fig. 7).

How the *Hira* gene is regulated is unknown. *Mtrr*gt/gt embryos at E10.5 (derived from *Mtrr*gt/gt intercrosses), which demonstrate a greater phenotypic risk than the *Mtrr*+/gt maternal grand-parental pedigrees[3], also displayed dysregulation of *Hira* mRNA and lncRNA expression (Supplementary Figs. 8k–m, 11d). This finding was in association with normal DNA methylation at the *Hira* DMR in *Mtrr*gt/gt embryos at E10.5 (Supplementary Fig. 8b), implicating additional mechanisms of epigenetic regulation. Histone methylation profiles at the *Hira* locus in normal ESCs and TSCs indicate a potential role for the *Hira* promoter and *Hira* DMR (Fig. 5a) in gene regulation that will require future investigation.

HIRA protein levels were also dysregulated in *Mtrr*gt/gt embryos and F2 wildtype embryos and placentas (Fig. 6e–h, Supplementary Fig. 11e, f). The pattern of dysregulation did not always occur in a manner predicted by the direction of *Hira* mRNA expression. For example, there was an increase in HIRA protein levels in F2 wildtype embryos when *Hira* mRNA was downregulated (Fig. 6b, d–h). This discrepancy might result from defective HIRA protein degradation, drastic translational upregulation of HIRA protein to compensate for low mRNA levels, or alternatively, negative feedback to down-regulate *Hira* mRNA owing to high levels of HIRA protein in the embryo. Further work will be required to delineate whether the HIRA chaperone mediates maternal phenotypic inheritance in the *Mtrr*gt model and other models of TEI.

## Discussion

We investigated potential mechanisms contributing to epigenetic inheritance in *Mtrr*gt mice, a unique model of mammalian TEI[3]. In the *Mtrr*gt model, inheritance of developmental phenotypes and epigenetic instability occurs via the maternal grandparental lineage with an F0 *Mtrr*+/gt male or female initiating the TEI effect[3]. Here, we assessed DNA methylation in spermatozoa to understand how the germline epigenome was affected by the *Mtrr*gt allele. We chose sperm due to its experimental tractability and to avoid the confounding factors of the F0 uterine environment when assessing epigenetic inheritance via the germline. We identified several distinct DMRs in regions of predicted importance in transcriptional regulation and epigenetic inheritance including regions of nucleosome retention and reprogramming resistance. This result illustrates widespread epigenetic instability in the male germline of the *Mtrr*gt model, particularly in the F0 *Mtrr*+/gt males of the maternal grandfather pedigree. While largely resolved in somatic tissue of subsequent wildtype generations, some germline DMRs were correlated with transcriptional changes at associated loci in the F1–F3 progeny. This proposed transcriptional memory of a germline DMR persisted for at least three generations, longer than previously reported in another model[4]. This observation indicates additional epigenetic factors beyond DNA methylation in the mechanism of TEI. Furthermore, the histone chaperone gene *Hira* emerged as a transcriptional biomarker and potential mediator of maternal phenotypic inheritance.

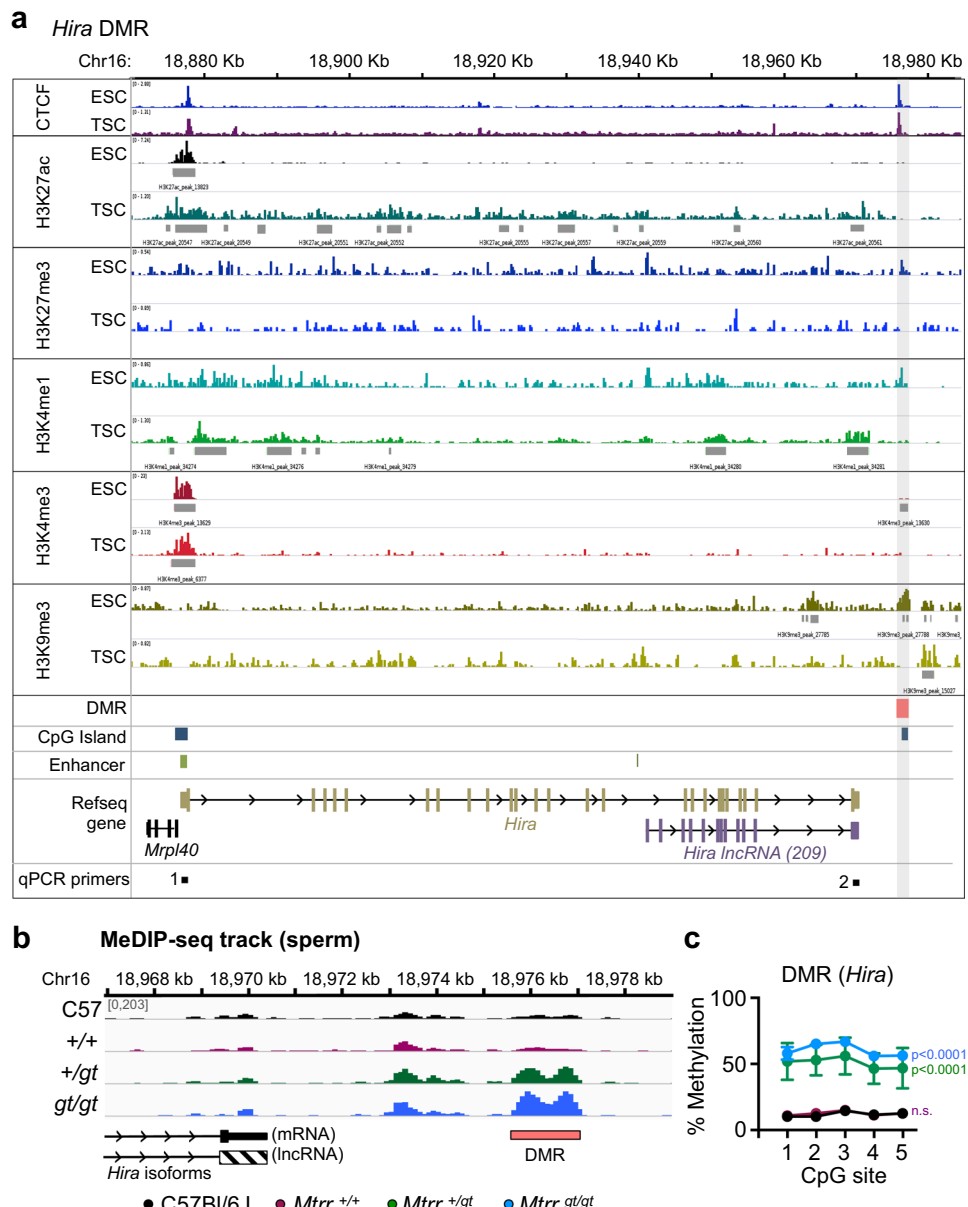

**Fig. 5 Epigenetic characteristics of *Hira* DMR in mice. a** Enrichment of DNA binding proteins (CCTC-binding factor, CTCF) and histone modifications (H3K27ac, H3K27me3, H3K4me1, H3K4me3, H3K9me3) in the *Hira* locus on mouse chromosome (Chr) 16 using published data sets[35] of chromatin immunoprecipitation followed by sequencing (ChIP-seq) analyses in cultured wildtype embryonic stem cells (ESCs) and trophoblast stem cells (TSCs). Dark grey rectangles indicate enrichment peak calling for each histone modification. Pink rectangle and light grey shading indicate the *Hira* differentially methylated region (DMR) identified in sperm of *Mtrr*^+/gt and *Mtrr*^gt/gt males. Blue rectangles indicate CpG islands. Green rectangles indicate enhancers. Schematics of protein-encoding (brown) and long non-coding RNA (lncRNA) encoding (purple) *Hira* isoforms are shown. Region of qPCR primer sets 1 and 2 are also indicated (black boxes). **b** Partial schematic drawing of *Hira* transcripts and *Hira* DMR (pink rectangle) in relation to MeDIP-seq reads in sperm from C57Bl/6 J (black), wildtype (*Mtrr*^+/+; purple), *Mtrr*^+/gt (green) and *Mtrr*^gt/gt (blue) males. *N* = 8 males/group. **c** The average percentage methylation at individual CpG sites (mean ± standard deviation) in the *Hira* DMR in sperm from C57Bl/6 J males (black circles; *N* = 4 males), *Mtrr*^+/+ males (purple circles; *N* = 4 males), *Mtrr*^+/gt males (green circles; *N* = 8 males) and *Mtrr*^gt/gt males (blue circles; *N* = 8 males). Two-way ANOVA with Sidak's multiple comparisons test performed on mean methylation per CpG site. *p* values are indicated for each male genotype compared to C57Bl/6 J and were similar for all CpG sites assessed within one genotype as indicated: C57Bl/6 J versus *Mtrr*^+/+, not significant (n.s.); C57Bl/6 J versus *Mtrr*^+/gt males or *Mtrr*^gt/gt males, *p* < 0.0001. Source data are provided as a Source Data file.

The extent to which genetic and epigenetic factors interact in this and other TEI models is unclear. One-carbon metabolism is involved in thymidine synthesis[54], and DNA breaks triggered by folate deficiency-induced uracil misincorporation were demonstrated in erythrocytes of splenectomised patients[55] and prostate adenoma cells[56]. However, WGS of *Mtrr*^gt/gt embryos excluded genetic instability in the *Mtrr*^gt mouse line because de novo mutations occurred at an expected[28] and comparable frequency to control embryos. The WGS data also discounted alternative phenotype-causing mutations outside of the *Mtrr* locus and, when compared with our sperm methylome data set, showed that differential CpG methylation was unlikely due to underlying genetic variation in the *Mtrr*^gt mouse line. Therefore, the epigenetic consequences of the *Mtrr*^gt allele rather than genetic

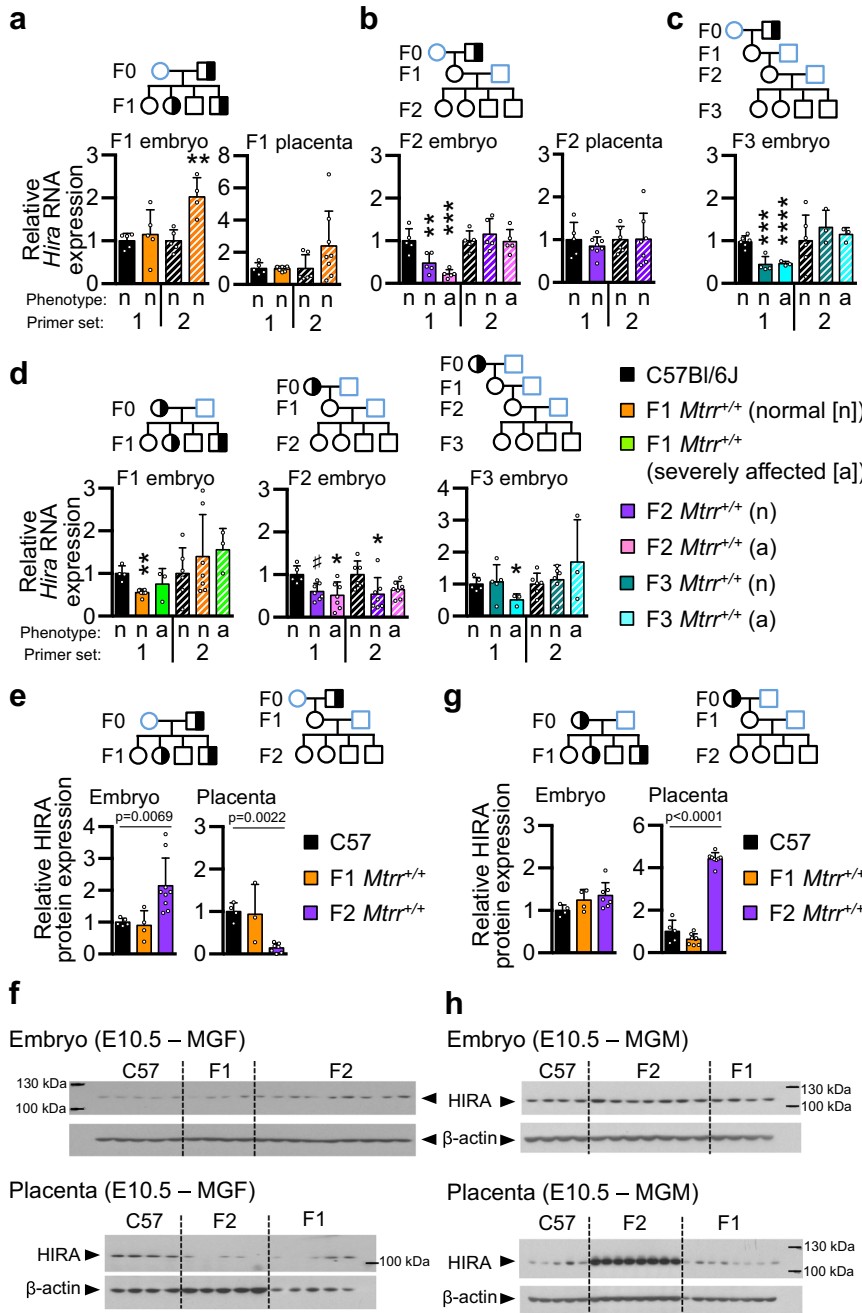

instability are more likely to instigate TEI in this model. Generating alternative *Mtrr* mutations and/or backcrossing the *Mtrr^gt* allele into a different mouse strain will further assess whether genetics has a role in TEI.

Our data show that inheritance of sperm DMRs by offspring somatic tissue was unlikely, as have other studies[4,49,50]. Instead, somatic cell lineages might inherit germline epigenetic instability in a broader sense. For instance, despite normal DNA methylation in F1–F3 wildtype embryos and placentas (E10.5) at the genomic locations highlighted by sperm DMRs, epigenetic instability associated with gene misexpression is still evident in F1 and F2 wildtype placentas at several loci[3]. It is possible that abnormalities in the sperm epigenome of F0 *Mtrr^+/gt* males might be reprogrammed and then stochastically and abnormally re-established/maintained in other genomic regions in wildtype offspring. This hypothesis might explain inter-individual phenotypic variability in the F2–F3 generations. However, we showed

that sperm of *Mtrr^+/+* males exhibited several DMRs that overlapped with sperm DMRs in *Mtrr^+/gt* males (representing their fathers). Therefore, reconstruction of specific atypical F0 germline methylation patterns in the F1 wildtype germline is possible in the *Mtrr^gt* model. Vinclozolin toxicant exposure model of TEI shows dissimilar DMRs in spermatozoa of F1 and F3 offspring[8], which suggests that epigenetic patterns might shift as the generational distance from the F0 individual increases.

Dietary folate deficiency causes differential methylation in sperm and craniofacial defects in the immediate offspring[49], though whether it leads to TEI is unknown. There was no overlap of DMRs when sperm methylomes were compared between the diet model and *Mtrr^+/gt* males. This result disputes the existence of folate-sensitive epigenomic hotspots in sperm. However, the severity of insult or technical differences (e.g., MeDIP-array[49] versus MeDIP-seq) might explain the discrepancy. Why only one *Mtrr^gt* allele sufficiently initiates TEI is unknown since a direct

**Fig. 6 Dysregulation of *Hira* RNA expression in embryos aligns with a pattern of maternal phenotypic inheritance. a–d** RT-qPCR analysis of *Hira* mRNA (solid bars, primer set 1) and *Hira* lncRNA (striped bars, primer set 2) expression in embryos and placentas at E10.5. The tissue and pedigrees were assessed (see also Supplementary Fig. 1a, d–e for the breeding scheme): **a** F1 wildtype (*Mtrr*$^{+/+}$) conceptuses from F0 *Mtrr*$^{+/gt}$ males (orange bars; *N* = 4–5 embryos, *N* = 8 placentas), **b** F2 wildtype conceptuses from F0 *Mtrr*$^{+/gt}$ males (purple bars, *N* = 5 embryos and *N* = 7–8 placentas from phenotypically normal (n) conceptuses; pink bars, *N* = 5 severely affected (a) embryos), **c** F3 wildtype conceptuses from F0 *Mtrr*$^{+/gt}$ males (teal bars, *N* = 3–4 phenotypically normal embryos; turquoise bars; *N* = 3 severely affected embryos), and **d** F1 wildtype embryos from F0 *Mtrr*$^{+/gt}$ females (orange bars, *N* = 7 phenotypically normal embryos; green bars, *N* = 3 severely affected embryos), F2 wildtype embryos from F0 *Mtrr*$^{+/gt}$ females (purple bars, *N* = 7 phenotypically normal embryos; pink bars, *N* = 7 severely affected embryos), and F3 wildtype embryos from F0 *Mtrr*$^{+/gt}$ females (teal bars, *N* = 5–7 phenotypically normal embryos; turquoise bars, *N* = 3 severely affected embryos). C57Bl/6 J conceptuses were controls (black bars, *N* = 4–8 embryos or placenta/experiment). **e–h** Western blot analysis showing HIRA protein expression in F1 wildtype (orange bars) and F2 wildtype (purple bars) conceptuses derived from **e** to **f** F0 *Mtrr*$^{+/gt}$ males or **g–h** F0 *Mtrr*$^{+/gt}$ females. C57Bl/6 J (C57) conceptuses (black bars) were assessed as controls. **e, f** C57Bl6/J: *N* = 4–5 embryos, *N* = 4–5 placentas; F1: *N* = 4 embryos, *N* = 3–8 placentas; F2: *N* = 7–9 embryos, *N* = 7–8 placentas. Images of western blot gels showing HIRA protein with β-actin as a loading control in **f** and **h** were quantified in **e** and **g**, respectively, using the background subtraction method. HIRA protein levels were normalised to β-actin. All RNA and protein data were plotted as mean ± standard deviation and relative to C57Bl/6 J (normalised to 1). Experiments were conducted in technical duplicates (protein) or triplicates (RNA). Statistical analyses: **a–d** Two-tailed independent *t* test or Kruskal–Wallis test with Dunn's multiple comparison. **a** **$p$ = 0.0034; **b** **$p$ = 0.0049, ***$p$ = 0.0002; **c** ***$p$ = 0.0004, ****$p$ < 0.0001; **d** F1: **$p$ = 0.0056, F2 (primer set 1): #$p$ = 0.06, *$p$ = 0.0203, F2 (primer set 2): *$p$ = 0.0457, F3: *$p$ = 0.0161. **e, f** One-way ANOVA. *p* values are shown on graphs. Pedigree legend: circle, female; square, male; blue outline, C57Bl/6 J line; black outline, *Mtrr*$^{gt}$ mouse line; white fill, *Mtrr*$^{+/+}$; half-black-half-white fill, *Mtrr*$^{+/gt}$; black fill, *Mtrr*$^{gt/gt}$. F0, parental generation; F1, first filial generation; F2, second filial generation; F3, third filial generation. Source data are provided as a Source Data file.

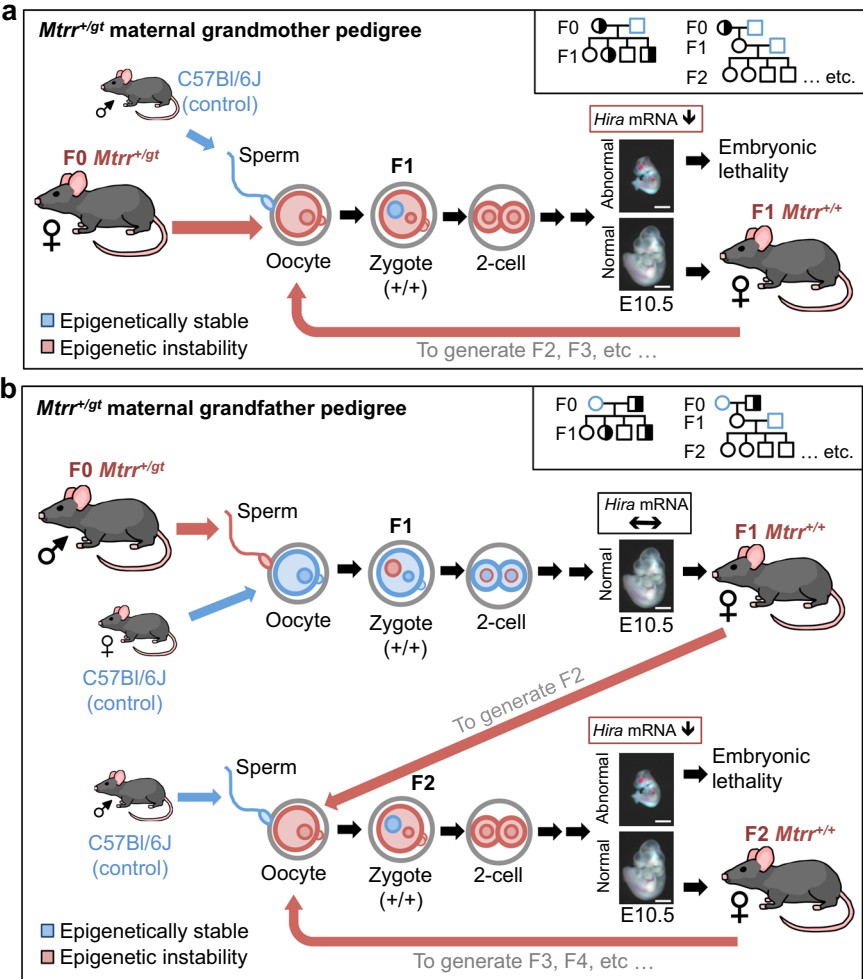

**Fig. 7 Proposed model of maternal grandparental phenotype inheritance in *Mtrr*$^{gt}$ model that implicates epigenetic instability in the germline.** A model proposing how epigenetic instability generated by the *Mtrr*$^{gt}$ mutation might be differently inherited over multiple generations depending upon whether TEI is initiated by **a** an oocyte or **b** sperm of an F0 *Mtrr*$^{+/gt}$ individual. Trends of *Hira* mRNA expression are shown. Pedigree legend: circle, female; square, male; blue outline, C57Bl/6 J control; black outline, *Mtrr*$^{gt}$ mouse line; white filled, *Mtrr*$^{+/+}$; half-white/half-black filled, *Mtrr*$^{+/gt}$. See also Supplementary Fig. 1 for breeding schemes. Scale bar: 500 μm. F0, parental generation; F1, first filial generation; F2, second filial generation; F3, third filial generation.

paramutation effect was not evident[3] and $Mtrr^{+/gt}$ mice do not display similar metabolic derangement to $Mtrr^{gt/gt}$ mice[3,12].

Whether specific DNA methylation patterns observed in F0 sperm of the maternal grandfather pedigree are reconstructed[57,58] in F1 oocytes is yet-to-be determined. Extensive differences between sperm and oocytes methylomes[59] will make this difficult to resolve and yet also emphasizes that additional epigenetic mechanisms are likely involved, such as histone modifications[10], sncRNA expression[1,9] and/or changes in nucleosome composition and spacing that alter nuclear architecture[59]. Several of these mechanisms implicate regulators like HIRA[25–27] (see below). Future studies will explore the extent to which differential epigenetic marks in oocytes from F0 $Mtrr^{+/gt}$ females are reconstructed in oocytes of subsequent generations, through embryo transfer experiments will be required to exclude the confounding effects of the F0 uterine environment[60].

Several recent studies have assessed sncRNA expression (e.g., microRNA or tRNA fragments [tsRNA])[1,2,9] in sperm as a mechanism in the direct epigenetic inheritance of disease. For example, manipulating paternal diet in mice is sufficient to alter tsRNA content in spermatozoa leading to altered expression of genes associated with MERVL elements in early F1 embryos[9] and metabolic disease in F1 adult offspring[1]. It is possible that sncRNAs in sperm from F0 $Mtrr^{+/gt}$ males are misexpressed and/or abnormally modified[61], and that this might contribute to TEI mechanisms. Although it is currently unclear how sperm ncRNA content causes phenotypic inheritance beyond the F1 generation, genes involved in the development of the primordial germ cell population within the F1 wildtype female embryo might be among those affected. Exploring sncRNA content in germ cells of the $Mtrr^{gt}$ model will help to better understand this question.

Several sperm DMRs in $Mtrr^{+/gt}$ males (e.g., $Cwc27$, $Tshz3$, $Hira$) were associated with transcriptional changes in F2–F3 wildtype embryos and adult liver. We focused our analysis on the $Hira$ DMR due to its responsiveness to the severity of the $Mtrr^{gt}$ genotype, the fact that $Hira^{-/-}$ embryos[52] phenocopy embryos in the $Mtrr^{gt}$ model[3], and because the potential consequences of HIRA dysfunction are multifaceted and implicated in epigenetic inheritance. The HIRA histone chaperone complex regulates histone deposition/recycling required to maintain chromatin integrity in the oocyte and zygote[25,26,62]. Additionally, HIRA is implicated in rRNA transcription[27]. Therefore, dysregulation of HIRA expression and/or function in the $Mtrr^{gt}$ model might alter nucleosome spacing and ribosome heterogeneity[63,64] with substantial implications for regulation of transcriptional pathways during germ cell and embryo development.

Since the F1 progeny differ phenotypically when derived from an F0 $Mtrr^{+/gt}$ male versus F0 $Mtrr^{+/gt}$ female[3], the mechanism initiating TEI potentially differs between sperm and oocytes. The extent of mechanistic overlap is not well understood. Certainly, paternally inherited epigenetic factors[1,2,4,9,10,49,61,65] are better studied than maternally-inherited factors[60,66]. The difference in phenotypic severity in the $Mtrr^{gt}$ model might relate to a cytoplasmically inherited factor in the F1 wildtype zygote (Fig. 7). HIRA is a suitable candidate since maternal HIRA[25,67] is involved in protamine replacement by histones in the paternal pronucleus[27]. Therefore, abnormal HIRA expression or function in oocytes would potentially perpetuate epigenetic instability in the next generation. In the case of the $Mtrr^{gt}$ model, dysregulation of $Hira$ and/or other maternal factors in oocytes might have a greater impact on epigenetic integrity in the early embryo than when dysregulated in sperm.

Overall, we show the potential long-term transcriptional and phenotypic impact of abnormal folate metabolism on germline DNA methylation and emphasise the complexity of epigenetic

mechanisms involved in TEI. Ultimately, our data indicate the importance of normal folate metabolism in both women and men of reproductive age for healthy pregnancies in their daughters and granddaughters.

## Methods

**Ethics statement**. This research was regulated under the Animals (Scientific Procedures) Act 1986 Amendment Regulations 2012 following ethical review by the University of Cambridge Animal Welfare and Ethical Review Body.

**Animal model**. $Mtrr^{Gt(XG334)Byg}$ (MGI:3526159) mouse line, referred to as the $Mtrr^{gt}$ model, was generated when a $\beta$-$geo$ gene-trap (gt) vector was inserted into intron 9 of the $Mtrr$ gene in 129P2Ola/Hsd ESCs[3]. $Mtrr^{gt}$ ESCs were injected into C57Bl/6 J blastocysts. Upon germline transmission, the $Mtrr^{gt}$ allele was back-crossed into the C57Bl/6 J genetic background for at least eight generations[3]. $Mtrr^{+/+}$ and $Mtrr^{+/gt}$ mice were from $Mtrr^{+/gt}$ intercrosses (Supplementary Fig. 1b). $Mtrr^{gt/gt}$ mice were produced from $Mtrr^{gt/gt}$ intercrosses (Supplementary Fig. 1c). C57Bl/6 J mice from The Jackson Laboratories (www.jaxmice.jax.org) and 129P2Ola/Hsd from Envigo (previously Harlan Laboratories [www.envigo.com]) were used as controls and were bred in house and separately from the $Mtrr^{gt}$ mouse line. Mice were housed in a temperature- and humidity-controlled environment with a 12 h light–dark cycle. All mice were fed a normal chow diet (Rodent No. 3 chow, Special Diet Services) ad libitum from weaning, which included (per kg of diet): 1.6 g choline, 2.73 mg folic acid, 26.8 µg vitamin B$_{12}$, 3.4 g methionine, 51.3 mg zinc. Mice were killed via cervical dislocation. Genotyping for the $Mtrr^+$ and $Mtrr^{gt}$ alleles and for sex ($Rbm31$) was performed[3,68,69] using PCR on DNA extracted from ear tissue or yolk sac using primer sequences in Supplementary Table 8.

To determine the multigenerational effects of the $Mtrr^{gt}$ allele in the maternal grandfather, the following mouse pedigree was established (Supplementary Fig. 1d). For the F1 generation, F0 $Mtrr^{+/gt}$ males were mated with C57Bl/6 J females and the resulting $Mtrr^{+/+}$ progeny were analysed. For the F2 generation, F1 $Mtrr^{+/+}$ females were mated with C57Bl/6 J males and the resulting $Mtrr^{+/+}$ progeny were analysed. For the F3 generation, F2 $Mtrr^{+/+}$ females were mated with C57Bl/6 J males and the resulting $Mtrr^{+/+}$ progeny was analysed. A similar pedigree was established to assess the effects of the $Mtrr^{gt}$ allele in the maternal grandmother with the exception of the F0 generation, which involved the mating of an $Mtrr^{+/gt}$ female with a C57Bl/6 J male (Supplementary Fig. 1e).

**Tissue dissection and phenotyping**. For embryo and placenta collection, timed matings were established and noon on the day that the vaginal plug was detected was considered embryonic day (E) 0.5. Embryos and placentas were dissected at E10.5 in cold 1× phosphate-buffered saline and were scored for phenotypes (see below), photographed, weighed, and snap frozen in liquid nitrogen for storage at −80 °C. All conceptuses were dissected using a Zeiss SteREO Discovery V8 microscope with an AxioCam MRc5 camera (Carl Zeiss). Livers were collected from pregnant female mice (gestational day 10.5), weighed and snap frozen in liquid nitrogen for storage at −80 °C. Both male and female conceptuses at E10.5 were assessed, as no sexual dimorphism is apparent at this stage[70]. While many individual mice were assessed over the course of this study, some of the individual tissue samples were assessed for the expression of multiple genes or for methylation at multiple DMRs.

A rigorous phenotyping regime was performed at E10.5 as previously described[3]. In brief, all conceptuses were scored for one or more congenital malformation (as appropriate for the developmental stage) including failure of the neural tube to close in the cranial or spinal cord region, malformed branchial arches, pericardial edema, reversed heart looping, enlarged heart, and/or eccentric chorioallantoic attachment. Twinning or haemorrhaging was also scored as a severe abnormality. Conceptuses with severe abnormalities were categorised separately from resorptions, the latter of which consisted of maternal decidua surrounding an indistinguishable mass of fetally derived tissue. Resorptions were not assessed in this study as they represented dead conceptuses. Embryos with <30 somite pairs were considered developmentally delayed. Embryos with 30–39 somite pairs but a crown-rump length more than two standard deviations (sd) from the mean crown-rump length of C57Bl/6 J control embryos were considered growth restricted or growth enhanced. Conceptuses were considered phenotypically normal if they were absent of congenital malformations, had 30–39 somite pairs, and had crown-rump lengths within two sd of controls. AxioVision 4.7.2 imaging software was used to measure crown-rump lengths (Carl Zeiss). Conceptus size at E10.5 was unaffected by litter size in all $Mtrr^{gt}$ pedigrees assessed[70].

**Spermatozoa collection**. Spermatozoa from cauda epididymides and vas deferens were collected from 16 to 20 week-old fertile mice using a swim-up procedure as previously described[38] with the following amendments. Spermatozoa were released for 20 min at 37 °C in Donners Medium (25 mM NaHCO3, 20 mg/ml bovine serum albumin,, 1 mM sodium pyruvate and 0.53% (vol/vol) sodium dl-lactate in Donners stock (135 mM NaCl, 5 mM KCl, 1 mM MgSO4, 2 mM CaCl$_2$ and 30 mM HEPES)). Samples were centrifuged at 500 × $g$ (21 °C) for 10 min. The supernatant was transferred and centrifuged at 1300 × $g$ (4 °C) for 15 min. After the majority of

supernatant was discarded, the samples were centrifuged at $1300 \times g$ (4 °C) for 5 min. Further supernatant was discarded and the remaining spermatozoa were centrifuged at $12,000 \times g$ for 1 min and stored at −80 °C.

**Nucleic acid extraction**. For embryo, trophoblast and liver tissue, genomic DNA (gDNA) was extracted using DNeasy Blood and Tissue kit (Qiagen) according to the manufacturer's instructions. RNA was extracted from tissues using the AllPrep DNA/RNA Mini Kit (Qiagen). For sperm, Solution A (75 mM NaCl pH 8; 25 mM EDTA) and Solution B (10 mM Tris-HCl pH 8; 10 mM EDTA; 1% SDS; 80 mM DTT) were added to the samples followed by RNAse A incubation (37 °C, 1 h) and Proteinase K incubation (55 °C, overnight) as was previously described[4]. DNA was extracted using phenol/chloroform/isoamyl alcohol mix (25:24:1) (Sigma-Aldrich) as per the manufacturer's instructions and pelleted using standard methods in TE buffer. DNA quality and quantity were confirmed using gel electrophoresis and QuantiFluor dsDNA Sample kit (Promega) as per the manufacturer's instructions.

**Whole-genome sequencing**. Non-degraded gDNA from two whole C57Bl/6 J embryos at E10.5 (one male, one female) and six whole $Mtrr^{gt/gt}$ embryos with congenital malformations at E10.5 (four males, two females) derived from $Mtrr^{gt/gt}$ intercrosses (Supplementary Fig. 1a, c) was sent to BGI (Hong Kong) for library preparation and sequencing. The libraries of each embryo were sequenced separately. Sequencing was performed with 150 bp paired-end reads on an Illumina HiSeq X machine. Quality control of reads assessed with FastQC (version 0.11.5, http://www.bioinformatics.babraham.ac.uk/projects/fastqc/). Adaptors and low-quality bases removed using Trim Galore (version 0.6.4, https://www.bioinformatics.babraham.ac.uk/projects/trim galore/). Summary metrics were created across all samples using the MultiQC package (version 1.4, http://multiqc.info)[71]. FastQ files were merges using seqkit (version 0.8.0)[72]. Sequencing reads were aligned to the C57Bl/6 J reference genome (GRCm38, mm10) using BowTie2 with default parameters (version 2.3.4, http://bowtie-bio.sourceforge.net/bowtie2/index.shtml)[73]. Duplicates were marked using Picard (version 2.9.0, http://broadinstitute.github.io/picard).

SV analysis was performed using Manta (version 0.29.6)[74]. SV were filtered using vcftools (version 0.1.15)[75]. In order to identify SNPs, the data were remapped to the $mm10$ reference mouse genome using BWA (version 0.7.15-r1144- dirty)[76]. Reads were locally realigned and SNPs and short indels identified using GenomeAnalysisTK (GATK, version 3.7)[77]. Homozygous variants were called when >90% of reads at the locus supported the variant call, whereas variants with at least 30% of reads supporting the variant calls were classified as heterozygous. Two rounds of filtering of variants were performed as follows. First, low-quality and biased variant calls were removed. Second, variants with: (i) simple repeats with a periodicity <9 bp, (ii) homopolymer repeats >8 bp, (iii) dinucleotide repeats >14 bp, (iv) low mapping quality (<40), (v) overlapping annotated repeats or segmental duplications, and (vi) >3 heterozygous variants fell within a 10 kb region were removed using vcftools (version 0.1.15) and bcftools (version 1.3.1) as was previously described[78]. The 129P2/OlaHsd mouse genome variation data were downloaded from Mouse Genomes Project[79]. The functional effect of SNPs was predicted using snpEff (version 4.3t)[80].

**Methylated DNA immunoprecipitation and sequencing**. MeDIP-seq[81] was carried out using 3 µg of sperm gDNA that was sonicated using a Diagenode Bioruptor UCD-200 to yield 200–700 bp fragments that were end-repaired and dA-tailed. Illumina adaptors for paired-end sequencing were ligated using the NEB Next DNA Library Prep Master Mix for Illumina kit (New England Biolabs). After each step, the DNA was washed using Agencourt AMPure XP SPRI beads (Beckman Coulter). IPs were performed in triplicate using 500 ng of DNA per sample, 1.25 µl of mouse anti-human 5mC antibody (0.1 mg/0.1 mL, clone 33D3; Eurogentec Ltd., Cat No. BI-MECY, RRID:AB_2616058), and 10 µl of Dynabeads coupled with M-280 sheep anti-mouse IgG bead (Invitrogen). The three IPs were pooled and purified using MinElute PCR Purification columns (Qiagen). Libraries were amplified by PCR (12 cycles) using Phusion High-Fidelity PCR Master Mix and adaptor-specific iPCR tag primers (Supplementary Table 9), and purified using Agencourt AMPure XP SPRI beads. The efficiency of the IP was verified using qPCR to compare the enrichment for DNA regions of known methylation status (e.g., methylated in sperm: $H19$ and $Peg3$ ICR, Supplementary Fig. 5a)[49] in the pre-amplification input and the IP fractions. MeDIP library DNA concentrations were estimated using the Kapa Library Quantification kit (Kapa Biosystems) and were further verified by running on an Agilent High Sensitivity DNA chip on an Agilent 2100 BioAnalyzer. Sequencing of MeDIP libraries was performed using 100 bp paired-end reads on an Illumina HiSeq platform at the Babraham Institute Next Generation Sequencing Facility (Cambridge, UK).

Quality assessment of the sequencing reads was performed using FastQC (version 0.11.5, http://www.bioinformatics.babraham.ac.uk/projects/fastqc/). Adaptor trimming was performed using Trim Galore (version 0.6.4, http://www.bioinformatics.babraham.ac.uk/projects/trim galore/). Reads were mapped to the GRCm38 (mm10) reference genome using Bowtie2 (version 2.3.4, http://bowtie-bio.sourceforge.net/bowtie2/index.shtml)[73]. All programmes were run with default settings unless otherwise stated. Sample clustering was assessed using principle component analysis, using the 500 most variable windows with respect to read coverage (as a proxy for methylation) for 5 kb window across all samples. Further

data quality checks and differential methylation analysis was performed using the MEDIPS package in R (version 1.40.0, using R version 3.4.2)[33]. The following key parameters were defined: BSgenome = BSgenome.Mmusculus.UCSC.mm10, uniq = 1e-3, extend = 300, ws = 500, shift=0. DMRs were defined as windows (500 bp) in which there was at least 1.5-fold difference in methylation (reads per kilobase million mapped reads (RPKM)) between C57Bl/6 J and $Mtrr$ sperm methylation level with a $p$ value <0.01. Adjacent windows were merged using BEDTools (version 2.27.0)[82]. The background methylome was defined as all 500 bp windows across the genome at which the sum of the average RPKM per genotype group was >1.0. The genomic localisations of DMRs including association with coding/non-coding regions and CpG islands were determined using annotation downloaded from University of California, Santa Cruz (UCSC)[83]. The percentage of DMRs associated with repetitive regions of the genome was calculated using RepeatMasker software (version, http://www.repeatmasker.org).

**Enrichment analysis of published ChIP-seq and ATAC-seq data**. Mean enrichment of specific histone modifications and THSS, CTCF, H3.3 and PRM1 binding in the DMR regions were determined using published data sets including ChIP-seq data in CD1 spermatozoa collected from cauda epididymis[42], ESCs[35] and TSCs[35], and ATAC-seq data in CD1 spermatozoa collected from cauda epididymis[42] and B6D2F1 epiblast and extraembryonic ectoderm at E6.5[43]. The source and accession numbers of processed ChIP-seq and ATAC-seq wig/bigwig files are shown in Supplementary Table 10 and accessible on GitHub (https://github.com/CTR-BFX/Blake_Watson). To ensure that the analysis was consistent across public data sets, all wig files were converted to bigwig using UCSC tools "wigToBigWig -clip" (http://hgdownload.soe.ucsc.edu/admin/exe/).

DMRs identified in sperm from all three $Mtrr$ genotypes (MeDIP-seq analysis) were combined to generate a list of 893 DMRs. To prevent the inclusion of DMRs associated with genomic variation, the 20 Mb region surrounding the $Mtrr$ gene (Chr13:58060780-80060780) was identified as 129P2Ola/Hsd genomic sequence was masked. This resulted in 459 DMRs for subsequent analysis. The ChIP-seq and ATAC-seq files in sperm[42] were originally aligned to mouse reference genome mm9. The files were converted to mouse reference genome mm10 to match the other published data sets in this analysis using LiftOver with the mm9ToMm10.over.chain (http://genome.ucsc.edu/cgi-bin/hgLiftOver). To identify the baseline enrichment profiles around the DMRs for specific histone modifications, THSS, CTCF, H3.3 or PRM1, a similar number of genomic regions were randomly selected using bedtools (v2.26.0)[82] with the following command: "bedtools shuffle -i DMRs.bed -g Mus_GRCm38_chrall.fa.fai -chrom -seed 27442958 -noOverlapping -excl Mtrr_mask20Mb.bed". The DMR profiles were created using deeptools (version 2.3.1)[84], via computeMatix 3 kb scaled windows, flanking regions of 6 kb, and a bin size of 200 bp, and plotted with plotProfile.

**Enhancer analysis**. To determine whether genetic variants or DMRs overlapped with known enhancer regions, FANTOM5 enhancer database[85] for GRCm38 mouse genome (https://fantom.gsc.riken.jp/5/; F5.mm10.enhancers.bed.gz) was used. All.bed files were applied to UCSC Genome Browser[83] to check for region-specific features. The distance in base pairs between DMRs and genes or between enhancers and genes was calculated using the closest coordinates (the start/end of DMR or enhancer) to the transcriptional start site (TSS) minus the TSS and then plus 1. To determine further interactions between enhancers and promoters, we analysed public Hi-C data sets[35] for ESC and TSC (E-MTAB-6585; ESC_promoter-other_interactions_GOTHiC.txt and

TSC_promoter-other_interactions_GOTHiC.txt) using WashU Epigenome Browser[86].

**Quantitative reverse transcription PCR (RT-qPCR)**. For RNA expression analysis, cDNA was synthesised using RevertAid H Minus reverse transcriptase (Thermo Scientific) and random hexamer primers (Thermo Scientific) using 1–2 µg of RNA in a 20-µl reaction according to manufacturer's instructions. PCR amplification was conducted using MESA Green qPCR MasterMix Plus for SYBR Assay (Eurogentec Ltd.) on a DNA Engine Opticon2 thermocycler (BioRad). The following cycling conditions were used: 95 °C for 10 min, 40 cycles: 95 °C for 30 sec, 60 °C for 1 min, followed by melt curve analysis. Transcript levels were normalised to $Hprt$ and/or $Gapdh$ RNA levels. Relative cDNA expression levels were analysed using the 2(-Delta Delta C(T)) method[87]. Experiments were conducted in technical triplicate with biological replicates indicated in the figure legends. Transcript levels in C57Bl/6 J tissue were normalised to 1. For primer sequences and concentrations, refer to Supplementary Table 8.

**Bisulfite mutagenesis and pyrosequencing**. Between 250 ng and 2 µg of gDNA extracted from each tissue sample was bisulfite treated with an Imprint DNA Modification Kit (Sigma). Control samples lacking DNA template were run to ensure that there was no contamination during the bisulfite conversion of DNA. To quantify DMR CpG methylation, pyrosequencing was performed. In all, 50 ng of bisulfite-converted DNA was used as a template for PCR, together with 0.2 µM of each biotinylated primer and 0.25 units of HotStarTaq PlusDNA Polymerase (Qiagen). Refer to Supplementary Table 11 for primer sequences, which were designed using PyroMark Assay Design Software 2.0 (Qiagen). PCR was performed in triplicate

using the following conditions: 95 °C for 5 min, 40 cycles of 94 °C for 30 sec, 56 °C for 30 sec, 72 °C for 55 sec and then 72 °C for 5 min. PCR products were purified using Strepdavidin Sepharose High Performance beads (GE healthcare). The beads bound to DNA were washed in 70% ethanol, 0.4 M NaOH and 10 mM Tris-acetated (pH 7.6) and then hybridized to the sequencing primer in PyroMark annealing buffer (Qiagen) according to the manufacturer's instructions. Pyrosequencing was conducted using PyroMark Gold reagents kit (Qiagen) on a PyroMark MD pyrosequencer (Biotage). The mean CpG methylation was calculated using three to eight biological replicates and at least two technical replicates. Analysis of methylation status was performed using Pyro Q-CpG software (version 1.0.9, Biotage).

**Mass spectrometry**. Sperm gDNA was digested into individual nucleoside components using the DNA Degradase Plus kit (Zymo Research) according to the manufacturer's instructions. The heat inactivation step was omitted. In all, 100 ng of degraded DNA per individual was sent to the Babraham Institute Mass Spectrometry Facility (Cambridge, UK), where global cytosine, 5mC and 5hmC was determined by liquid chromatography-tandem mass spectrometry (LC-MS/MS) as previously described[88]. The sperm of nine males were assessed per genotype and analysed in pools each containing three unique individuals. All pooled samples were analysed in triplicate. Global 5mC and 5hmC levels are reported as percentages relative to C.

**Western blotting**. Embryos and placentas at E10.5 were homogenised in lysis buffer (20 mM Tris [pH 7.5], 150 mM NaCl, 1 mM ethylenediaminetetraacetic acid (EDTA), 1 mM ethylene glycol-bis(β-aminoethyl ether)-N,N,N′,N′-tetraacetic acid, 1% Triton X-100, 2.5 mM sodium pyrophosphate, 1 mM β-glycerolphosphate, 1 mM $Na_3VO_4$ and complete mini EDTA-free proteases inhibitor cocktail [Roche Diagnostics]) with Lysing Matrix D ceramic beads (MP Biomedical) using a MagNA Lyser (Roche Diagnostics) at 5500 rpm for 20 sec. Samples were incubated on ice for 5 min and then homogenised again at 5500 rpm for 20 sec. Homogenates were then incubated on ice for 20 min with brief intervening vortexing steps occurring every 5 min. Samples were then centrifuged at $10,000 \times g$ for 5 min. Supernatant from each sample was transferred to a new tube and centrifuged again at $10,000 \times g$ for 5 min to ensure that all residual tissue was removed. The protein concentration of tissue lysates was determined using bicinchoninic acid (Sigma-Aldrich). Proteins were denatured with gel loading buffer (50 mM Tris [pH 6.8], 100 mN DTT, 2% SDS, 10% glycerol and a trace amount of bromophenol blue) at 70 °C for 10 min. Equivalent amounts of protein were resolved by 8–10% SDS–PAGE and blotted onto nitrocellulose (0.2 μm, Amersham Protran) with a semi-dry blotter (GE Healthcare). The membrane was stained with Ponceau S solution (Sigma-Aldrich) and the resulting scanned image was used as a loading control. After washing, the membrane was blotted with 5% skimmed milk in Tris-buffered saline containing 0.1% Tween-20 (TBS-T) before incubation with 1:1000 dilution of monoclonal rabbit anti-human HIRA (clone D2A5E, Cell Signalling Technology, cat. No. 13307, RRID:AB_2798177) overnight at 4 °C or 1:10,000 dilution of monoclonal mouse anti-human β-actin (clone AC-74, Sigma-Aldrich, Cat. No. A2228, RRID:AB_476997) for 1 h at room temperature. Primary antibodies were diluted in TBS-T. The membrane was incubated with 1:10,000 dilution of donkey anti-rabbit IgG conjugated to horseradish peroxidase (HRP; GE Healthcare, cat. No. NA934, RRID:AB_772206) diluted in 2.5% skimmed milk in TBS-T or 1:10,000 dilution of sheep anti-mouse IgG conjugated to HRP (GE Healthcare, cat. No. NA931V, RRID:AB_772210) diluted in TBS-T. The signal of resolved protein was visualized by Amersham enhanced chemiluminescence (ECL) Western Blotting Analysis System (GE Healthcare) using Amersham Hyperfilm ECL (GE Healthcare). A flat-bed scanner (HP Scanjet G4050) was used to scan films. Experiments were conducted in technical duplicates with the biological replicates indicated in the figure legends. Band intensities were determined with background subtraction using ImageJ (64-bit) software (version 1.48; NIH, USA). Full gel blots are available in the Source Data file.

**Statistical analysis**. Statistical analysis was performed using GraphPad Prism software (version 7). RT-qPCR data were analysed by independent two-tailed unpaired $t$ tests or ordinary one-way ANOVA with Dunnett's or Sidak's multiple comparison testing. SV and SNP data were analysed by independent two-tailed unpaired $t$ tests with Welch's correction. Bisulfite pyrosequencing data, SV chromosome frequency and DMR distribution at repetitive elements were analysed by two-way ANOVAs with Dunnett's, Sidak's or Tukey's multiple comparisons tests. A binomial test (Wilson/Brown) was used to compare the observed frequency of nucleosome occupancy at DMRs to expected values. Western blot data were analysed using a non-parametric Kruskal–Wallis test with Dunn's multiple comparisons test. $p < 0.05$ was considered significant unless otherwise stated.

**Reporting summary**. Further information on research design is available in the Nature Research Reporting Summary linked to this article.

## Data availability

All relevant data are available from the corresponding author upon reasonable request. WGS data have been deposited in ArrayExpress database at EMBL-EBI under accession number E-MTAB-8513 and MeDIP-Seq data accession number is E-MTAB-8533. The Hi-C data[35] accession number is E-MTAB-6585. The source and accession numbers of processed ChIP-seq and ATAC-seq wig/bigwig files are listed in Supplementary Table 10. Source data are provided with this paper.

## Code availability

The in-house scripts used for the analysis can be found freely accessible in the following online repository: https://github.com/CTR-BFX/Blake_Watson and https://doi.org/10.5281/zenodo.3832249).

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

## Acknowledgements

We thank Drs Nozomi Takahashi and Tessa Bertozzi for their technical support, Dr. Claire Senner for critical discussion, and Drs Jamie Hackett and Miguel Branco for .bed files. The MeDIP-seq and LC-MS/MS were performed at the Babraham Institute (UK). WGS was performed by BGI (Hong Kong). This work was funded by grants from Lister Institute for Preventative Medicine and the Centre for Trophoblast Research (CTR) to E. D.W and from the MRC MR/J001597 and Wellcome Trust WT095606 (to A.C.F.-S). The following support was provided: Wellcome Trust 4-year DTP in Developmental Mechanisms (to G.E.T.B.) and CTR funding (to G.E.T.B., H.W.Y., X.Z., R.S.H.).

## Author contributions

E.D.W. conceived the project. G.E.T.B. collected sperm, performed DNA/RNA extractions, generated WGS and MeDIP libraries, and performed RT-qPCR and bisulfite pyrosequencing analyses. G.E.T.B. and E.D.W. dissected tissue and phenotyped conceptuses. G.E.T.B. and E.D.W. collected and analysed the data. X.Z., R.S.H. and G.E.T.B. designed and performed bioinformatics analyses. H.W.Y. performed the western blotting analysis. G.E.T.B., A.C.F.-S., G.J.B. and E.D.W. designed the experiments and interpreted the results. E.D.W. and G.E.T.B. wrote the manuscript. All authors read and revised the manuscript.

## Competing interests

The authors declare no competing interests.
