## [Peer Review File · Nature Communications]

REVIEWER COMMENTS

Reviewer #1 (Remarks to the Author):

The current study is an extension of the group's previous work (Padmanabhan, Cell 2013) in which they discovered that the Mtrr-gt mouse line (with reduced MTRR expression and disrupted folate metabolism) initiate a maternal transgenerational epigenetic inheritance (TEI) over multiple generations. A key feature of this model is that a male or female Mtrr+/gt mice is sufficient to cause TEI regarding developmental phenotypes in the genetically wildtype Mtrr+/+ offspring, which suggest epigenetic modifications on the "wildtype" germline. The current study set out to explore the underline mechanism of the TEI phenomenon in the Mtrr-gt model regarding 1) the genetic stability (DNA sequence), which turn out to be normal; 2) DNA methylation in the F0 sperm and somatic tissue of the offspring, showing that DNA methylation is altered in the sperm but these methylation changes were reprogrammed/restored in wildtype F1 and F2 generations; 3) the 'epigenetic memory' of sperm DMRs (differentially methylated regions) regarding the persistently altered transcriptional activity in spite of the restoration of the methylation level, showing that certain important genes, including Hira, remain abnormally expressed in the F1-F3 generations, which may account for the observed TEI of phenotypes. Although the question regarding how these abnormal gene expression are maintained across generations is not addressed and remain a major challenge of the field, I appreciate the comprehensive works performed by the group which have forward the field. I have a few additional advices listed below:

The current scenario of observed sperm DMRs getting disappeared in the offspring while the phenotype still get transmitted across generation is resonating with the team's another work using a diet-induced model (Radford, Science 2014); and also represented in a more recent work of cigarette smoke exposure in which the exposure induce sperm DMRs and offspring phenotype, but the sperm DMRs do not overlap with the DMRs in the offspring somatic tissue (PLoS Genet. 2020 Jun 10;16(6):e1008756.); these evidence enhanced the concept that DNA methylation changes in sperm are not directly inherited. These observations could be discussed integratively and from which setup the stage to explore other epigenetic information carriers that are being involved. Indeed, the current study have looked into the histone marks by Chip-seq in using existing datasets from sperm, ESCs and TSCs. And have shown that in ESCs and sperm, there are quite specific enrichment of K4me3, K27ac and THSS at the Hira DMR (Figure.5). This is interesting observation and suggest an interaction between these marks, could it be that the Hira gene activity is controlled by these marks instead of DNA methylation? Although there is no related Chip-data available using the current Mtrr-gt model, this deserves more detailed discuss (regarding K4me3, K27ac and THSS) and this may open future directions for studying the current model.

An additional point the paper should consider is the involvement of RNA-mediated effects in the current model. It would be interesting to explore the sperm non-coding RNA profile in the Mtrr-gt model - I know it is beyond the score of the present paper, but there are several lines of clues that worth to be discussed. Especially, Hira is essential for rRNA transcription, ribosomal protein nuclear translocation and ribosome assembly (Dev Cell. 2014 Aug 11; 30(3): 268–279.). This is a potentially important clue as it has been recently proposed that sperm RNA mediated ribosome heterogeneity in early embryo might generates translational specificity to define the metabolic phenotype of the offspring (Nat Rev Endocrinol. 2019 Aug;15(8):489-498.). Also, rRNA-derived small RNAs (rsRNAs) has been extensively discovered in both mouse and human sperm and these sperm rsRNAs are sensitive to dietary changes (Nat Cell Biol. 2018 May;20(5):535-540; PLoS Biol. 2019 Dec

26;17(12):e3000559.) possibly contributing to epigenetic inheritance. Finally, there is evidence showing that rDNA methylation at specific loci is sensitive to environmental exposure (Science. 2016 Jul 29;353(6298):495-8.), whether this can relate to rRNA transcription and rRNA level remains unexplored but it looks possible. The discovery of Hira in the current paper looks very interesting and may serve as a further clue that link rRNA transcription/ribosome function for epigenetic inheritance. I strongly recommend the authors to discuss these pieces of evidence in the context and to bring new insights.

Reviewer #2 (Remarks to the Author):

The manuscript "Disruption of folate metabolism causes germline epigenetic instability and distinguishes HIRA as a biomarker of maternal transgenerational epigenetic inheritance" by Blake et al. follow up on their previous investigation using an *Mtrr* hypomorphic mutation in mouse as an animal model of TEI. This work further describes this TEI model system showing that disruption in an F0 generation leads to transient DNA methylation defect in gametes. As with other studies of the same kind it is found that while a phenotype and associated gene expression changes (i.e. Hira expression) are transmitted through several generation, the DNA methylation changes are not. In short an epigenetic mark, DNA methylation is found to only partially correlates with TEI phenotype but no functional testing of its relevance to the phenotype is provided. As such the work is not very novel. Still the authors made a good attempt at ruling out a genetic basis to explain the phenotype and carry out a very detailed description of DNA methylation and gene expression changes.

The major weakness of the work comes from the fact that the authors characterize DNA methylation in the male germline when this phenotype is shown to be transmitted through the female germline. The authors point out that while the phenotype can be initiated from a male F0 it somehow skips F1 in that case. This is the only case when a phenotype is NOT induced in the next generation. Yet this the F0 germline that the authors use for the DNA methylation characterisation. The fact that performing DNA methylation in the female germline is more challenging is not convincing and a better case for the rationale/relevance of their choice should be put forward.

1/ Genetic changes analysis

-No genes in the 20Mb region around *Mtrr* gt allele have expression changes. Are there any of the SNP/SV located in known enhancers. If yes what about the genes regulated by such enhancers (could be away from the 20 Mb domain?)

-Not clear to me why stable expression of transposable element would be evidence of genomic stability? Is it because changes in sequence of transposable element are more difficult to assess because of their repeated nature hence expression is used as a proxy?

-Any evidence that the performed analysis would be able to spot genetic instability? How does the observed rate compare with rate of mutation reported elsewhere? Any possible positive control?

-suppl fig3c-d splice junction seem to show increased mutagenesis in gt background. What are the gene affected?

-Not clear how the SNP/SV calling is carried out (single embryo or pooled embryos data?). The authors mention de novo mutation rate. Can they evaluate if SNP/SV in same location between embryos?

2/ DNA methylation analysis

-DMRs are found to be in different location than SNP/SV. If DMRs are in enhancer, are they

contacting promoter with SNP/SV?

- fig 2b: A single heatmap representing the methylation profile of all DMRs identified in all 3 genotypes (versus the C57 ctrl) would highlight both the DMRs shared by the different genotypes and the DMRs specific to each genotypes.

-lines 170 to 180: What would be useful would be to show the 54 common DMR between genotype in fig 2f/g/h

-Statement 179-181' altogether...." Very vague

- Association with sperm nucleosome

A 1.94% nucleosome "retention rate" is indicated. It is not clear what this % entails (no details in M&M as far as I can tell). Need further explanation on this part for reader to fully interpret. Are authors using data from Peters's lab or Rando's. It is quite important to indicate the dataset used and why as these two groups use different chip protocol and have different conclusions as to where nucleosomes are retained along sperm genome.

Related to this issue line 196 authors indicate an enrichment for PRM1 while they state before that DMR are enriched for nucleosome. Since nucleosome are replaced by protamine in sperm there is a discrepancy here. Are they the same DMR or is the "double enrichment" indicative of heterogeneity in the sperm population? Maybe provide a heat map showing nucleosome and PRM1 occupancy in DMRs.

-Table 1 showing the results for the 54 common DMRs would be informative.

-Are DMRs (i.e in fig3) located in known enhancers?

-Supplfig 9-10/fig5a- difficult to tell binding from tracks> was peak calling performed? Show peak underneath track- Are tracks chip-inut?

3/gene expression analysis

This part of the manuscript focuses on Hira mRNA and protein level. The overall message seems to be that Hira misregulation tracks with phenotypic abnormality.

Based on figure 6 it seems to be true in the MGF lineage but not necessarily in the MGM lineage (i.e. abnormal embryo in F1, normal embryo in F3 , fig 6f). This could be seen as evidence that TEI initiated from MGF or MGM entails different mechanisms?

Very curious opposite trend between changes in mRNA level and changes in protein level. How quantitative is the WB analysis carried out?

Given the discrepancy between mRNA/protein levels trend, what would be the authors "marker" for TEI?

4/In abstract "exhibit memory of germline methylation defect" imply that the observed gene expression defect across generation are the results of the DNA methylation changes. I Would rephrase to indicate that the gene expression changes track with the phenotype while the DNAm are not.

Reviewer #3 (Remarks to the Author):

The current studies utilize the authors' Mtrr(gt) mutation mice that have reduced Mtrr expression, reduced MTR enzymatic activity, and altered folate metabolism to investigate inter- and trans-generational epigenetic changes. Mtrr(gt) male or female can pass on the effect via wt daughters for at least 4 generations in normal folate conditions, but this must start with initial female. The authors have previously shown using embryo transfer that some phenotypic alterations such as congenital malformation is independent of maternal uterine issues. Therefore, the authors have examined DNA methylation in placentas of F1 and F2 and shown changes as well, although what these altered methylation patterns mean is not

examined. The authors next examine sperm DNA methylation from F0 and inheritance of tissue methylation, and subsequently identify HIRA as a specific candidate for changes (important in chromatin stability).

Overall, the data are of high quality and the studies reflect an important next step in the authors' work. However, major concerns are noted as to the authors' ability to determine cause and effect, and the logic of examination of sperm methylation changes in a model where no phenotypic outcome is found related to these sperm changes. The paper seems to be a combination of 3 different stories that would be much stronger if broken into different papers. Minimally, the authors would need ICSI or IVF for their sperm study to provide causal inference and a connection between the sperm data and inheritance of the next generation of something more than methylation changes, again without demonstration of the impact or outcome of these changes the data remain less strong. There are additional concerns as to study design and data interpretation:

Questions:

1. Overall, missing causality and lack of logic in connection of some of the figures. The paper seems to be 3 separate papers: 1) mutation rate, 2) sperm, 3) mom and Hira. Figure 6 is especially unclear. Hira is an example of no causal changes, only weak associations.
2. Figure 2 – the sex of embryos is not listed. And the impact of litter effects is also not included. In many cases throughout the paper, the N's are quite small (N=3), so litter effects and clarity in embryo and placenta sex is essential.
3. In many cases, the authors have listed effects or outcomes by 'affected' or 'severely affected', which could potentially bias the results. How are the authors controlling for the bias of inclusion of severely affected offspring, when these are more likely to be closer to death, and hence any gene expression or changes being examined would seem questionable for the health of the cells. Again, this brings up the concern for distinguishing between cause and effect.
4. Alters differential placental DNA methylation – but what does this mean?
5. The authors are largely focused on examining transgenerational epigenetic inheritance – exclusively on DNA methylation. But what is missing is evidence linking this inheritance to any phenotype – how are these related in this paper? The authors need to distinguish between these two.
6. Figure 6 western blots - the authors need to show the bands.
7. Figure 6 data overall are quite perplexing and seemingly weak evidence to support any connection between the intergenerational effects with a lncRNA and Hira. The data do not show this pattern in many cases, and it is a bit of a stretch to infer causality. Further, there is no demonstration of any functional outcome for potential changes in Hira. Where is any histone functional change?
8. Is it really TEI if each subsequent generation requires an exposure to prior generation change in folate metabolism resulting from their development and intrauterine environment? Does the current data rule that out? This should be discussed.
9. Focus on Hira following the sperm methylation studies is awkward. Authors could develop this better. Fig 3 shows no change in DMR for Hira? Doesn't show up changed until Fig 4 for expression in the F2 embryo.
10. The introduction needs to explain the Mtrr-gt more clearly – as a reviewer, I had to search back in the literature to figure out what it was. As it is the major focus of the paper, reader shouldn't have to search for what it is or how it was made and validated. The authors could make a more clear explanation as to what the change in folate metabolism and what parent came from – maybe in a schematic? Getting lost in the weeds of genotype, but not clear what the authors hypothesis is related to exposure for germ cell and reproductive

pathway is.

11. The authors have previously shown that Mtrr +/- gt male mice have reduced homocysteine, not hyper-, how does that fit into this story? These males' sperm are being affected by their genotype and folate metabolism in seemingly very different, opposing ways.

12. For sperm assay methods, it appears no swim-up collection or assessment for motility was done. How are the authors confirming live cells or changes in sperm functionality? Has this already been examined – using IVF or ICSI?

13. Determination of 'abnormal' for embryo phenotype – if reproductive processes/implantation etc. are altered by genotype, does that change interpretation of outcomes for role of Mtrr? Authors should discuss fertility outcomes – is there an effect on rate of implantation or fertilization? Sex ratio of offspring with phenotype or abnormal outcome?

14. Why so much variance of penetrance here? Could there be an interactive effect of intrauterine position? Was intrauterine position controlled for in placenta or embryo studies here?

15. Seems to be a large literature of paternal studies missed in the authors' citations for many groups focusing on sperm RNAs and intergenerational inheritance.

Responses to reviewers' comments (Blake et al., NCOMMS-20-19078-T)

Reviewer 1 comments:

The current study is an extension of the group's previous work (Padmanabhan, Cell 2013) in which they discovered that the *Mtrr^{gt}* mouse line (with reduced MTRR expression and disrupted folate metabolism) initiate a maternal transgenerational epigenetic inheritance (TEI) over multiple generations. A key feature of this model is that a male or female *Mtrr^{+/gt}* mice is sufficient to cause TEI regarding developmental phenotypes in the genetically wildtype *Mtrr^{+/+}* offspring, which suggest epigenetic modifications on the "wildtype" germline.

The current study set out to explore the underline mechanism of the TEI phenomenon in the *Mtrr^{gt}* model regarding 1) the genetic stability (DNA sequence), which turn out to be normal; 2) DNA methylation in the F0 sperm and somatic tissue of the offspring, showing that DNA methylation is altered in the sperm but these methylation changes were reprogrammed/restored in wildtype F1 and F2 generations; 3) the 'epigenetic memory' of sperm DMRs (differentially methylated regions) regarding the persistently altered transcriptional activity in spite of the restoration of the methylation level, showing that certain important genes, including *Hira*, remain abnormally expressed in the F1-F3 generations, which may account for the observed TEI of phenotypes. Although the question regarding how these abnormal gene expression are maintained across generations is not addressed and remain a major challenge of the field, I appreciate the comprehensive works performed by the group which have forward the field. I have a few additional advices listed below:

1/ The current scenario of observed sperm DMRs getting disappeared in the offspring while the phenotype still get transmitted across generation is resonating with the team's another work using a diet-induced model (Radford, Science 2014); and also represented in a more recent work of cigarette smoke exposure in which the exposure induce sperm DMRs and offspring phenotype, but the sperm DMRs do not overlap with the DMRs in the offspring somatic tissue (PLoS Genet. 2020 Jun 10;16(6):e1008756.); these evidence enhanced the concept that DNA methylation changes in sperm are not directly inherited. These observations could be discussed integratively and from which setup the stage to explore other epigenetic information carriers that are being involved.

Authors' Response [1]: We thank the reviewer for bringing to our attention Murphy *et al.* (2020 *Plos Genetics*). By citing this study together with Radford *et al.* (2014, *Science*) and Lambrot *et al.* (2013 *Nat Commun*), we have now more clearly indicated in the revised manuscript (lines 257-60) that our findings showing the resolution sperm DMR in somatic tissue of the direct offspring are reminiscent of data in these other studies. We also better emphasize that alternative epigenetic mechanisms are likely important for phenotypic inheritance (lines 260-1).

2/ Indeed, the current study has looked into the histone marks by Chip-seq in using existing datasets from sperm, ESCs and TSCs. And have shown that in ESCs and sperm, there are quite specific enrichment of K4me3, K27ac and THSS at the *Hira* DMR (Figure 5). This is an interesting observation and suggests an interaction between these marks, could it be that the *Hira* gene activity is controlled by these marks instead of DNA methylation? Although there is no related Chip-data available using the current *Mtrr*^{gt} model, this deserves more detailed discuss (regarding K4me3, K27ac and THSS) and this may open future directions for studying the current model.

Authors' Response [2]: We preface this response by saying that our analysis of the *Hira*, *Cwc27* and *Tshz3* loci now includes peak calling for histone modifications in ESCs and TSCs only (see our response to Reviewer 2 [comment 2.8], new Fig 5a, Supplementary fig. 8-9 in revised manuscript). Since there was no input data for sperm, epiblast and extraembryonic ectoderm in the corresponding published data sets, similar peak calling could be performed and the associated data was removed from the revised manuscript. As a result, the conclusions were substantially clarified (see lines 273-86 in the revised manuscript).

We think that Reviewer 1's description above (i.e., H3K4me3, H3K27ac, and THSS peaks in ESCs and TSCs) was describing the *Hira* promoter rather than the *Hira* DMR (see Fig. 5a). Regardless, we agree with the Reviewer that the histone modification peaks at the *Hira* promoter in ESCs and TSCs are interesting and possibly influence *Hira* gene regulation in concert with the *Hira* DMR. We have now added a sentence in the revised manuscript (see lines 335-7) to emphasize this possibility and to highlight that a future direction will be to better understand *Hira* gene regulation.

3.1/ An additional point the paper should consider is the involvement of RNA-mediated effects in the current model. It would be interesting to explore the sperm non-coding RNA profile in the *Mtrr*^{gt} model - I know it is beyond the scope of the present paper, but there are several lines of clues that worth to be discussed.

Authors' response [3.1]: We agree with the Reviewer that RNA-mediated mechanisms of epigenetic inheritance are important to explore alongside DNA and histone modifications. We have now included more discussion about this point in the revised manuscript (lines 409-18). We are actively exploring RNA-mediated mechanisms in the lab with robust preliminary data showing differential expression of specific small non-coding RNA (sncRNA) species in sperm of F0 *Mtrr*^{+/gt} males. As the Reviewer would likely agree, this data requires its own manuscript to consider this concept in sufficient detail.

3.2/ Especially, HIRA is essential for rRNA transcription, ribosomal protein nuclear translocation and ribosome assembly (Dev Cell. 2014 Aug 11; 30(3): 268–279). This is a potentially important clue as it has been recently proposed that sperm RNA mediated ribosome heterogeneity in early embryo might generate translational specificity to define the metabolic phenotype of the offspring (Nat Rev Endocrinol. 2019 Aug;15(8):489–498). Also, rRNA-derived small RNAs (rsRNAs) has been

extensively discovered in both mouse and human sperm and these sperm rsRNAs are sensitive to dietary changes (Nat Cell Biol. 2018 May; 20(5):535–540; PLoS Biol. 2019 Dec 26;17(12):e3000559) possibly contributing to epigenetic inheritance. Finally, there is evidence showing that rDNA methylation at specific loci is sensitive to environmental exposure (Science. 2016 Jul 29;353(6298):495–8), whether this can relate to rRNA transcription and rsRNA level remains unexplored but it looks possible. The discovery of Hira in the current paper looks very interesting and may serve as a further clue that link rRNA transcription/ribosome function for epigenetic inheritance. I strongly recommend the authors to discuss these pieces of evidence in the context and to bring new insights.

Authors' response [3.2]: The potential role of ribosomal heterogeneity and selective mRNA translation in epigenetic inheritance is intriguing, particularly given that metabolism might potentially skew this process through differential expression of sncRNA (Zhang *et al*, 2019 *Nat Rev Endocrinol*; Kim *et al*, 2017 *Nature*). Indeed, the proposed role of HIRA in rRNA transcription, ribosomal protein nuclear translocation and ribosome assembly (Lin *et al.*, 2014 *Dev Cell*) will be interesting to explore in the *Mtrr^{gt}* model as part of the next chapter of this story when HIRA function is delved into more extensively in the context of TEI. We have now indicated in the revised manuscript the function of HIRA in ribosome assembly and function (see lines 291-4, 425-8 in the revised manuscript). We appreciate the importance of this hypothesis yet due to the restrictions on manuscript length and the speculative nature of this discussion in context of the *Mtrr^{gt}* model given the data currently presented, we have kept the discussion to a minimum.

As a matter of further insight into this hypothesis, our MeDIP-seq analysis indicated a hypermethylated sperm DMR upstream of *Rn45s* in *Mtrr^{+/gt}* and *Mtrr^{gt/gt}* males relative to C57Bl/6J controls (see below Fig. R1.1). *Rn45s* encodes for 45S pre-ribosomal RNA and occurs in rDNA repeats on several chromosomes, but are only annotated on Chr 17 of the reference genome. Therefore, the accuracy of the sperm *Rn45s* DMR data was in doubt and the data was not incorporated into our manuscript. Of further note, our WGS data set showed no rDNA copy number variation.

Fig. R1.1. Putative *Rn45s* sperm DMR. **a,b**, MeDIP-seq dataset in spermatozoa of C57Bl/6J, *Mtrr^{+/+}*, *Mtrr^{+/gt}* and *Mtrr^{gt/gt}* males revealed a putative DMR upstream of the *Rn45s* gene on Chr17 (N=8 males/group). **a**, Schematic of *Rn45s* DMR. **b**, MeDIP-seq track of the *Rn45s* locus in sperm from each experimental group.

Reviewer 2 comments:

The manuscript "Disruption of folate metabolism causes germline epigenetic instability and distinguishes HIRA as a biomarker of maternal transgenerational epigenetic inheritance" by Blake et al. follow up on their previous investigation using an *Mtrr* hypomorphic mutation in mouse as an animal model of TEI. This work further describes this TEI model system showing that disruption in an F0 generation leads to transient DNA methylation defect in gametes. As with other studies of the same kind it is found that while a phenotype and associated gene expression changes (i.e. *Hira* expression) are transmitted through several generation, the DNA methylation changes are not. In short an epigenetic mark, DNA methylation is found to only partially correlates with TEI phenotype but no functional testing of its relevance to the phenotype is provided. As such the work is not very novel. Still the authors made a good attempt at ruling out a genetic basis to explain the phenotype and carry out a very detailed description of DNA methylation and gene expression changes.

0.1/ The major weakness of the work comes from the fact that the authors characterize DNA methylation in the male germline when this phenotype is shown to be transmitted through the female germline. The authors point out that while the phenotype can be initiated from a male F0 it somehow skips F1 in that case. This is the only case when a phenotype is NOT induced in the next generation. Yet this the F0 germline that the authors use for the DNA methylation characterisation.

Author's response [0.1]: We respectfully disagree with the reviewer's statement that TEI of phenotypes in the *Mtrr^{gt}* model are only "shown to be transmitted through the female germline" and that characterizing the male germline is a "major weakness" of this study. We likely contributed to this misunderstanding by stating throughout the manuscript that the *Mtrr^{gt}* mouse line is "a model of maternal TEI". In reality, it is a model of maternal *grandparental* TEI, which includes the maternal grandfather (and the male germline). We clarified the text by changing 'maternal TEI' to 'maternal grandparental TEI' throughout the revised manuscript (see lines 18, 37, 59, 240).

We apologize that the original justification of assessing the male germline was unclear in the original manuscript. Since either maternal grandparent can initiate TEI as determined through analysis of robustly controlled genetic pedigrees (Padmanabhan et al, 2013 *Cell*), exploring the germline of F0 males is equally relevant to F0 females. Even though F1 *Mtrr^{+/+}* conceptuses derived from an F0 *Mtrr^{+/gt}* male do not display gross phenotypes at E10.5, there are clear indicators of direct epigenetic inheritance, and certainly TEI, that emphasizes the importance of the F0 male germline:

- i) F1 *Mtrr^{+/+}* conceptuses derived from an F0 *Mtrr^{+/gt}* male display substantial and functionally-relevant locus-specific differential DNA methylation that associates with gene misexpression at E10.5, even in the absence of gross phenotype at E10.5 (Padmanabhan et al, 2013 *Cell*).
- ii) While only one developmental time-point was assessed in the original study, we have since shown that adult F1 *Mtrr^{+/+}* females derived from F0 *Mtrr^{+/gt}* males display adult-onset hematopoiesis defects (Padmanabhan et al, 2018 *J Phys*).

iii) When considering a broader context, F1 *Mtrr*^{+/+} females also show defects in their uterine environment causing growth phenotypes in the F2 *Mtrr*^{+/+} grandprogeny and defects in their germ cells causing congenital malformations in the F2 *Mtrr*^{+/+} grandprogeny, even after embryo transfer (Padmanabhan et al, 2013 *Cell*). The latter implicates germline epigenetic inheritance initiated by F0 *Mtrr*^{+/gt} males. Similar phenotypes are observed in the F3 and F4 *Mtrr*^{+/+} generations further supporting the existence of TEI caused by F0 *Mtrr*^{+/gt} males (Padmanabhan et al, 2013 *Cell*).

Therefore, we argue that the absence of morphological phenotypes at E10.5 in the F1 generation should not preclude analysis of the F0 male germline (as suggested by both Reviewers 2 and 3) when it clearly plays an initiating role in direct epigenetic inheritance and TEI. Indeed, by assessing the male germline in the current study, we not only observed widespread disruption of DNA methylation due to the *Mtrr*^{gt} allele but also identified DMRs that showed a pattern of transcriptional memory at their associated genes that was only revealed by assessing both the *Mtrr*^{+/gt} maternal grandfather and maternal grandmother pedigrees. We have better justified our analysis of the male germline in the revised manuscript (see lines 59-73). See also below our response Reviewer 2, point 0.2.

0.2/ The fact that performing DNA methylation in the female germline is more challenging is not convincing and a better case for the rationale/relevance of their choice should be put forward.

Author's response [0.2]: We think that technical challenges, and the associated financial constraints, of assessing epigenetic inheritance in oocytes are sufficient justification for assessing F0 sperm over F0 oocytes for two reasons: i) More F0 female mice are required to collect sufficient numbers of oocytes (compared to F0 males for sperm), which increases housing and experimental costs. ii) The identification of true inherited epigenetic marks from F0 oocytes requires an embryo transfer experiment to separate the effects of maternal physiology of the F0 *Mtrr*^{+/gt} females from germline epigenetic inheritance. This experiment involves surgically transferring F1 blastocysts into the uteri of control recipient females prior to scoring for phenotypes. We previously showed via embryo transfer that the uterine environment of F1 *Mtrr*^{+/+} females was responsible for a proportion of the developmental phenotypes (i.e., fetal growth defects) observed in the F2 generation (Padmanabhan et al, 2013 *Cell*). Likewise, a similar effect is hypothesized for F0 females. This is a fairly substantial experimental undertaking with the need for additional controls given that the embryo transfer technique itself affects the epigenome and transcriptome (Menelaou et al, 2020 *Reprod*; Vrooman et al, 2020 *Development*).

Therefore, initially exploring the effects of the *Mtrr*^{gt} mutation on the male germline epigenome was logical and more streamline since we did not need to consider the F0 uterine environment in this situation. Our data in the current manuscript confirms that the germline epigenome is affected in the *Mtrr*^{gt} model, which provides justification to embark on the more technically challenging experiments on the oocyte epigenome over multiple generations in future studies. Beyond what is stated in our response to Reviewer 2 (point 0.1) above, we have further clarified our rationale behind focusing on

the male germline in the first instance (see lines 76-80, 242-4, 352-4, 405-8 in the revised manuscript).

1/ Genetic changes analysis

1.1/ No genes in the 20Mb region around *Mtrr^{gt}* allele have expression changes. Are there any of the SNP/SV located in known enhancers. If yes what about the genes regulated by such enhancers (could be away from the 20 Mb domain?)

Authors' response [1.1]: The only SV in the 20 Mb region was the gene-trap itself. To determine whether SNPs within the 20 Mb region surrounding the *Mtrr^{gt}* locus (Fig. 1a) overlapped with an enhancer region, we used FANTOM5 enhancer database for mouse (see lines 615-24 in the revised methods). SNPs that were considered in this analysis were identified in at least two (out of six) of the *Mtrr^{gt/gt}* embryos assessed and were absent in the controls. No SNPs were found to directly overlap a known enhancer in this region. We have highlighted this point in the revised manuscript (see lines 113-4). The nearest distance between a SNP and an enhancer was 2.5 kb. The enhancer in question was 1.5 kb from the transcriptional start site of *Nsun2*. *Nsun2* mRNA expression was rigorously assessed in this manuscript in several tissue types and displayed normal expression in all cases (Fig. 1d) indicating that a tentative SNP/enhancer interaction is unlikely to affect *Nsun* gene expression.

1.2/ Not clear to me why stable expression of transposable element would be evidence of genomic stability? Is it because changes in sequence of transposable element are more difficult to assess because of their repeated nature hence expression is used as a proxy?

Authors' response [1.2]: For clarity, our hypothesis is as follows: transposable elements (TEs) are suppressed in the host epigenome by DNA methylation to prevent them from jumping around the genome and causing mutation (Deniz et al, 2019 *Nat Rev Genet*). Since genome-wide patterns of DNA methylation are altered in the context of the *Mtrr^{gt/gt}* mutation (Padmanabhan et al, 2013 *Cell*; the current study), the hypothesis was that dysregulation of TEs might result leading to an increase in expression/activity, and thus increased risk for mutation. Therefore, we measured TE expression to indicate a lack of repression and potential for mutagenesis. A similar method was implemented in Kim et al, 2014 (*Mol Cell* 56(4): 564-79) to assess genetic integrity in the context of global DNA demethylation. Indeed, the repetitive nature of TEs makes it difficult resolve their relocation using our whole genome sequencing (WGS) data set. As a reminder, no change in TE expression was evident in *Mtrr^{gt/gt}* tissue (Fig. 1e), which supports the maintenance of genetic integrity. We have clarified our rationale for this experiment (see lines 115-7 in the revised manuscript).

1.3/ Any evidence that the performed analysis would be able to spot genetic instability? How does the observed rate compare with rate of mutation reported elsewhere? Any possible positive control?

Authors' response [1.3]: Published data on de novo mutation rates in mice is variable and sparse. Our WGS data analysis identified ~11,000 variants/embryo (regardless of

genotype) when compared to the C57Bl/6J reference genome. One study predicted a normal mutation rate in the mammalian genome as high as $\sim 2.2 \times 10^9$ base pairs/year (Kumar & Subramanian, 2002 *PNAS*). Using WGS, others identified fewer than 1,000 SNPs when comparing two littermates of the A^{vy} mouse line (Oey et al, 2015 *Epigenetics & Chromatin*). No littermates were used in our study, and so more variants were expected. A third study re-analyzed published WGS data sets to show that two closely related mouse strains (i.e., C57Bl/6J and C57Bl/6N) have a difference of $\sim 10,000$ putative variants (Simon et al, 2013 *Genome Biology*). The C57Bl/6J and $Mtrr^{gt/gt}$ embryos in our study were from robustly inbred mouse lines that are maintained separately. Therefore, the latter study is the closest available comparator to our analysis since the de novo mutation rate is in line with what we observe. We have included this point in revised manuscript (see lines 108).

Given the expense of WGS, we were unable to include tissue samples with known genetic instability as a positive control. Regardless, we argue that the 20 Mb region surrounding the *Mtrr* gene-trap insertion site (Fig. 1a) acts as an internal positive control. Relative to the reference genome, the WGS analysis identified a striking difference in the number of genetic variants in this 20 Mb region resulting from the maintained 129P20Ia/HSD genetic sequence in $Mtrr^{gt/gt}$ embryos when compared to a similar analysis in C57Bl/6J control embryos. For this reason, we are confident that our bioinformatics analysis was appropriate and capable of identifying and quantifying differences in genetic variants between the experimental groups. A statement was included in the revised manuscript (see lines 101-2).

1.4/ Suppl fig. 3c-d: splice junction seem to show increased mutagenesis in gt background. What are the gene affected?

Authors' response [1.4]: Three SVs were present in the splice junctions of two genes (*Ttc14* and *Rps6ka4*). However, none of these SVs were present in all six $Mtrr^{gt/gt}$ embryos assessed. Four (out of six) $Mtrr^{gt/gt}$ embryos showed SVs on chromosome 3 in the gene *Ttc14* (see Fig. R2.1 below), which encodes a scaffold protein known to mediate protein-protein interactions (Allan & Ratajczak, 2011 *Cell Stress Chaperones*). A knockout mouse model of *Ttc14* has not been characterised and the protein interactions associated with TTC14 are unknown. Only one $Mtrr^{gt/gt}$ embryo showed an SV in a splice junction of the gene *Rps6ka4*, which encodes for ribosomal protein S6 kinase, polypeptide 4 on chromosome 19 (see Fig. R2.2 below). *Rps6ka4*^{-/-} mice do not display a phenotype (Anaieva et al, 2008 *Nat Immunol*). None of these SVs overlapped with known enhancers or sperm DMRs. It is unclear whether these SVs alter splicing, and this might be interesting to follow up in future studies. Given the complexity of the current manuscript and the fact that these particular SVs were not present in all $Mtrr^{gt/gt}$ embryos, we have opted not to include this information in the revised manuscript.

Fig. R2.1. A screen shot of UCSC genome browser showing *Ttc14* gene on chromosome 3 and SVs identified in *Mtrr^{gt/gt}* embryos at splice junctions (highlighted in blue).

Fig. R2.2. A screen shot of UCSC genome browser showing *Rps6ka4* gene on chromosome 19 and the SV identified in one *Mtrr^{gt/gt}* embryo at a splice junction (highlighted in blue).

1.5/ Not clear how the SNP/SV calling is carried out (single embryo or pooled embryos data?). The authors mention de novo mutation rate. Can they evaluate if SNP/SV in same location between embryos?

Authors' response [1.5]: The genomes of single embryos were sequenced separately (i.e., not pooled). We apologize for this oversight, which has now been clarified on lines 91-2, 105, and 527-8 in the revised manuscript. Most of the SNPs and SVs appeared in different locations in each embryo, which we attributed to de novo mutations. Through an intersectional analysis, we identified four SVs and 21 SNPs that were present in all six *Mtrr^{gt/gt}* embryos and absent in C57Bl/6J embryos (see new Supplementary fig. 2e,f). Most of these variants were present in intergenic or intronic regions and did not overlap with an enhancer (see new Supplementary tables 1 and 2). The exception was a tandem duplication in *Mtrr^{gt/gt}* embryos representing a known copy number variant in the C57Bl/6J mouse strain (Watkins-Chow & Pavan, 2008 *Genome Res*). The associated text discussing the new data is on lines 108-10 of the revised manuscript.

2/ DNA methylation analysis

2.1/ DMRs are found to be in different location than SNP/SV. If DMRs are in enhancer, are they contacting promoter with SNP/SV?

Authors' response [2.1]: Only eight sperm DMRs overlapped with known enhancers, which are now shown in a new Supplementary table 4. We used Hi-C data sets in ESCs and TSCs (Schoenfelder et al, 2018 *Nat Comm*; E-MTAB-6585) to show that none of these DMR/enhancer regions were associated with promoters that included a SNP or SV in a developmental context (see lines 615-24 in revised Methods section). A statement was included in the text of the revised manuscript (see lines 173-5). See also our response to Reviewer 2, point 2.7 for more information about DMRs and enhancers.

2.2/ Fig 2b: A single heatmap representing the methylation profile of all DMRs identified in all 3 genotypes (versus the C57 ctrl) would highlight both the DMRs shared by the different genotypes and the DMRs specific to each genotypes.

Authors' response: A heatmap showing common, paired and unique DMRs between the three different *Mtrr* genotypes (relative to controls) was generated and is now shown in a new Fig. 2c. The associated text is shown on lines 174-6 of the revised manuscript.

2.3/ Lines 170-180: What would be useful would be to show the 54 common DMR between genotype in fig 2f/g/h.

Authors' response [2.3]: The genomic location of the common DMRs was analysed. The data revealed that this subset of DMRs showed a similar distribution as sperm DMRs found in *Mtrr^{+/+}* and *Mtrr^{+/gt}* males. This new data is presented in a new Supplementary fig. 5d-e in a format similar to the total DMR data, as requested by the Reviewer (see also new Fig. 2e-g [formerly Fig. 2f-h], and lines 194-7 in the revised manuscript). As per Reviewer 2, point 2.6, we generated a new Supplementary table 5

listing the features of the 54 common sperm DMRs between *Mtrr*^{+/+}, *Mtrr*^{+/gt} and *Mtrr*^{gt/gt} males (cited on line 175 of revised manuscript).

2.4/ Statement 179-181' altogether..." Very vague

Authors' response [2.4]: "Altogether, these results might have implications for gene regulation." has been replaced with "While the majority of sperm DMRs were located within CpG deserts, a proportion of DMRs from *Mtrr*^{+/gt} and *Mtrr*^{gt/gt} males were enriched in CpG islands ($p < 0.0014$, Chi-squared; Fig. 2g), **which has implications for gene regulatory control.**" (See lines 192-5 of revised manuscript).

2.5.1/ Association with sperm nucleosome. A 1.94% nucleosome "retention rate" is indicated. It is not clear what this % entails (no details in M&M as far as I can tell). Need further explanation on this part for reader to fully interpret. Are authors using data from Peters's lab or Rando's. It is quite important to indicate the dataset used and why as these two groups use different chip protocol and have different conclusions as to where nucleosomes are retained along sperm genome.

Authors' response [2.5.1]: We apologize to the Reviewer for the lack of clarity in this section of the manuscript, some of which was admittedly erroneous. We have now indicated that the nucleosome retention rate in mouse spermatozoa is ~1%, which was previously described by the Peters' lab in Hisano *et al.* (2013 *Nat Protoc*) (see lines 198-9 of revised manuscript). Additionally, we have more clearly stated (on lines 199-201) that published datasets showing nucleosome retention in sperm differ with respect to regional distribution and frequency (Erkek *et al.* 2013 *Nat Struct Mol Biol*; Carone *et al.* 2014 *Dev Cell*). The value of 1.94% represents the *expected* percentage of DMRs within nucleosome retention sites. This was determined using the MNase-seq dataset of C57Bl/6J sperm in Erkek *et al.* (Peters' lab) whereby 10,000 randomly selected genomic regions of 500 bp (as a proxy for DMRs) were assessed for nucleosome overlap. The *observed* frequency of overlap between sperm DMRs from *Mtrr*^{gt} males and nucleosome retention sites was 14.5-34.1% of DMRs, which was significantly higher than the expected value listed above ($p < 0.0001$, binomial test). We have clarified the text of the revised manuscript (see lines 198-208 and Table 1).

2.5.2/ Related to this issue line 196 authors indicate an enrichment for PRM1 while they state before that DMR are enriched for nucleosome. Since nucleosome are replaced by protamine in sperm there is a discrepancy here. Are they the same DMR or is the "double enrichment" indicative of heterogeneity in the sperm population? Maybe provide a heat map showing nucleosome and PRM1 occupancy in DMRs.

Authors' response [2.5.2]: We thank the reviewer for pointing out this apparent discrepancy between nucleosome retention and protamine (PRM1) enrichment. Only 14.5-34.1% of DMRs overlap with retained nucleosomes in sperm (depending on male genotype; see Table 1 and lines 204-206 of the revised manuscript). This means that the substantial number of remaining DMRs do not overlap with sites of nucleosome retention and instead are expected to associate with protamine-enriched regions.

Therefore, there is heterogeneity within the wider sperm DMRs group with respect to genomic location. We have changed the text to account this discrepancy (see lines 218-9).

2.6/ Table 1 showing the results for the 54 common DMRs would be informative.

Authors' response [2.6]: A new Supplementary Table 5 was generated to show information about the common DMRs among the three *Mtrr* male genotypes. This table is cited on line 175 of the revised manuscript. See also our responses to Reviewer 2, points 2.2 and 2.3 for more analysis on the common DMRs.

2.7/ Are DMRs (i.e in fig 3) located in known enhancers?

Authors' response [2.7]: As indicated in our response to Reviewer 2, point 2.1, there were only eight DMRs that overlapped with enhancers. Of the DMRs assessed in Fig 3, only the *Cwc27* DMR overlapped with an enhancer (see below in Fig. R2.3). We updated Fig. 3c and Supplementary fig. 7i to schematically depict this enhancer/DMR overlap and we updated Supplementary fig. 8 to show the location of the enhancer with respect to the *Cwc27* DMR and histone modifications peaks. These data further support a role for the *Cwc27* DMR in transcriptional regulation. The text of the revised manuscript was updated on lines 276-7, 283-6.

Fig. R2.3. A screen shot of UCSC genome browser showing sperm DMRs identified in *Mtrr*^{+/-} and *Mtrr*^{-/-} males (region highlighted in blue) located within *Cwc27* gene. A known enhancer overlaps with the DMR as determined using FANTOM5 software. See also Fig. 3c, Supplementary table 4, Supplementary figs. 7i and 8.

2.8/ Supplementary figs. 9 and 10/ Fig. 5a: Difficult to tell binding from tracks> was peak calling performed? Show peak underneath track- Are tracks chip-inut?

Authors' response [2.8]: We thank the reviewer for this comment. Based on the data available, we were able to perform peak calling for the ChIP-seq data sets in embryonic stem cells (ESCs) and trophoblast stem cells (TSCs). This is now clearly shown in new Fig. 5a (*Hira* DMR), Supplementary figs. 8 (*Cwc27* DMR) and 9 (*Tshz3* DMR) of the revised manuscript. These data have revealed that the DMRs in ESCs were enriched for one or more repressive mark, while the DMRs in TSCs were absent of histone marks. Since no input data was available for the ChIP-seq and ATAC-seq data sets in spermatozoa, epiblast and extraembryonic ectoderm, we removed these tracks from the analysis. This omission clarifies the conclusions of this section of the manuscript. We have changed the text accordingly on lines 278-86 in the revised manuscript.

3/Gene expression analysis

3.1/ This part of the manuscript focuses on *Hira* mRNA and protein level. The overall message seems to be that *Hira* misregulation tracks with phenotypic abnormality. Based on figure 6 it seems to be true in the MGF lineage but not necessarily in the MGM lineage (i.e. abnormal embryo in F1, normal embryo in F3, fig 6f). This could be seen as evidence that TEI initiated from MGF or MGM entails different mechanisms?

Authors' response [3.1]: Based this comment and several comments from Reviewer 3, we have clarified the text associated with an updated Fig. 6 (see lines 288-346, new Fig. 6 and Supplementary Fig. 10). The main conclusion remains the same: *Hira* RNA is misexpressed in a manner that indicates transcriptional memory of the sperm DMR up to the F3 wildtype generation and that suggests it is a biomarker of maternal phenotypic inheritance. The latter conclusion is supported by *Hira* mRNA misexpression only in generations that derive from an oocyte with an *Mtrr*^{+/*gt*} ancestor (never sperm), which was determined by assessing two different genetic pedigrees.

We agree with the reviewer that different mechanisms are likely involved when an F0 *Mtrr*^{+/*gt*} male versus an F0 *Mtrr*^{+/*gt*} female initiates TEI. We tried to convey this in Fig. 7 of the original manuscript, which we think is still relevant and have kept it in the revised manuscript with edited associated text (see lines 327-30, 429-38). In the *Mtrr*^{gt} pedigree, epigenetic instability in the oocyte or maternally-derived factors that control epigenetic stability in the zygote (e.g., HIRA) might cause more severe phenotypic effects in the offspring than epigenetic instability in the sperm or paternally-inherited factors in the zygote. It is possible that a common inherited epigenetic factor to sperm and oocyte (e.g., altered sncRNA content, DNA methylation, or histone modifications) might cause TEI but that its presence in the oocyte might have more profound effects on epigenetic stability in the zygote. Maternally-derived factors that cause TEI are not well studied and we hope that our current study with shed some light onto the matter.

3.2/ Very curious opposite trend between changes in mRNA level and changes in protein level. How quantitative is the WB analysis carried out?

Authors' response [3.2]: To quantify the western blot data, we performed the standard protocol of background subtraction for HIRA bands normalized to β -actin using ImageJ (64-bit) software (highlighted in the Methods section, lines 681-2). This was followed by

a non-parametric Kruskal-Wallis test with Dunn's multiple comparison test. We agree that the trend was unexpected and we hypothesize three possibilities for this result: i) HIRA translation was drastically up-regulated to compensate for low *Hira* transcript expression, ii) *Hira* mRNA expression was down-regulated in response to high HIRA protein levels in a negative feedback loop, or iii) protein degradation pathways are altered in *Mtrr* embryos. Following from the latter hypothesis, we have preliminary western blot data suggesting that total ubiquitinated protein content is increased in F2 *Mtrr*^{+/-} embryos but not F1 *Mtrr*^{+/-} embryos from the *Mtrr*^{+gt} maternal grandfather pedigree compared to C57Bl/6J control embryos at E10.5 (see below in Fig. R2.4). This observation supports an accumulation of HIRA protein based on a proteasome pathway defect. Whether HIRA protein is among the ubiquitinated protein pool is unclear, but it may be worth pursuing experimentally in future studies. We have included a statement in the revised manuscript highlighting potential reasons behind the mismatch between *Hira* mRNA and HIRA protein levels (lines 340-3).

Fig. R2.4. Increased ubiquitinated protein content in F2 *Mtrr*^{+/-} embryos derived from an *Mtrr*^{+gt} maternal grandfather compared to C57Bl/6J controls. **a**, Relative ubiquitinated (Ub) protein levels in F1 and F2 *Mtrr*^{+/-} embryos at E10.5 derived from an F0 *Mtrr*^{+gt} male compared to C57Bl/6J controls as determined by western blot. Data shown as mean ± sd, relative to C57Bl/6J levels, which were normalized to 1. (N=4-9 embryos per experimental group). One-way ANOVA, *p<0.05. **b**, Western blot using antibody against ubiquitin. Ponceau stain is also shown in lower panel.

3.3/ Given the discrepancy between mRNA/protein levels trend, what would be the authors "marker" for TEI?

Authors' response [3.3]: We apologize that the conclusion was unclear in the original manuscript. We propose that the transcriptional down-regulation of *Hira* mRNA expression is a marker of maternal phenotypic inheritance. Based on this comment and comments from Reviewer 3, the text in this section has been substantially edited for clarity (lines 24-7, 299-346, 360-64, 439 of the revised manuscript).

4/ In abstract "exhibit memory of germline methylation defect" imply that the observed gene expression defect across generation are the results of the DNA methylation changes. I would rephrase to indicate that the gene expression changes track with the phenotype while the DNAm are not.

Authors' response [4]: As suggested, we rephrased the statement in the abstract to focus on transcriptional changes that associate with phenotypic inheritance rather than methylation inheritance (see lines 24-7).

Reviewer 3 comments:

The current studies utilize the authors' *Mtrr(gt)* mutation mice that have reduced *Mtrr* expression, reduced MTR enzymatic activity, and altered folate metabolism to investigate inter- and trans-generational epigenetic changes. *Mtrr(gt)* male or female can pass on the effect via wt daughters for at least 4 generations in normal folate conditions, but this must start with initial female. The authors have previously shown using embryo transfer that some phenotypic alterations such as congenital malformation is independent of maternal uterine issues. Therefore, the authors have examined DNA methylation in placentas of F1 and F2 and shown changes as well, although what these altered methylation patterns mean is not examined. The authors next examine sperm DNA methylation from F0 and inheritance of tissue methylation, and subsequently identify HIRA as a specific candidate for changes (important in chromatin stability).

0.1/ Overall, the data are of high quality and the studies reflect an important next step in the authors' work. However, major concerns are noted as to the authors' ability to determine cause and effect, and the logic of examination of sperm methylation changes in a model where no phenotypic outcome is found related to these sperm changes. The paper seems to be a combination of 3 different stories that would be much stronger if broken into different papers. Minimally, the authors would need ICSI or IVF for their sperm study to provide causal inference and a connection between the sperm data and inheritance of the next generation of something more than methylation changes, again without demonstration of the impact or outcome of these changes the data remain less strong.

Authors' response [0.1]: We strongly disagree with the Reviewer's assessment that our model does not show cause and effect, and that no phenotypic outcome is found in relation to 'sperm changes'. Additionally, we apologise that the rationale for assessing sperm was unclear. These two points were similar concerns of Reviewer 2, and we kindly refer Reviewer 3 to our detailed responses above (see Authors' responses to Reviewer 2's concerns [0.1] and [0.2], pages 4-5 of this report). We agree with the Reviewer 3 that there is a lot of data in this manuscript. We have gone through the revised manuscript to ensure that the common narrative (i.e., assessing potential mechanisms of TEI in the *Mtrr^{gt}* model) is more obvious and that the sections of the manuscript are better linked.

We value the Reviewer's suggestion of ICSI or IVF experiments. However, we think that these experiments are outside of the scope of this manuscript. We are currently undergoing these types of experiments in the lab to better understand how germ cell nuclear and cytoplasmic components are affected by the *Mtrr^{gt}* mutation and whether one or both components are required to cause TEI of phenotypes. This will be the next chapter in the *Mtrr^{gt}* story. We emphasize that our highly-controlled genetic pedigree analysis in combination with embryo transfer (Padmanabhan et al, 2013 *Cell*) sufficiently shows that the *Mtrr^{+/gt}* mutation in male mice alters their sperm leading to defects in their *Mtrr^{+/+}* daughters' germ cells to cause developmental defects in their *Mtrr^{+/+}* grandprogeny, great grandprogeny, and great, great grandprogeny (cause and effect).

Very little is known about the mechanisms of TEI and no one has yet shown a specific molecule or epigenetic factor that is responsible for phenotypic inheritance over multiple generations. Certainly, sncRNA content in sperm is a good candidate because of its role in direct epigenetic inheritance, which links it to metabolic disease in the direct offspring (Chen et al, 2016 *Science*). How sncRNA or other epigenetic mechanisms cause phenotypic inheritance for more than two generations is the ‘million dollar question’ and we think that the Reviewer holds us at too high a standard. The importance of this study is that we characterize the genetic and epigenetic landscape in a well-established model of TEI and use our data to make key hypotheses about how epigenetic instability might be passed between generations. We show potential transcriptional memory of several sperm DMRs up to three generations later (further than has been previously shown in mammals), which occurs in a manner that predicts maternal phenotypic inheritance as supported by direct evidence in genetic pedigrees. No maternally-inherited factors are known in mammalian TEI models. Our data provides information about novel candidates (e.g., HIRA), and highlights the mechanistic complexities of TEI and the difficulties of experimentally teasing apart these factors. Our manuscript further emphasizes that the *Mtrr^{gt}* model is will be key to addressing the epigenetic mechanisms behind TEI.

There are additional concerns as to study design and data interpretation:

Questions:

1. Overall, missing causality and lack of logic in connection of some of the figures. The paper seems to be 3 separate papers: 1) mutation rate, 2) sperm, 3) mom and Hira. Figure 6 is especially unclear. Hira is an example of no causal changes, only weak associations.

Authors’ response [1]: We have addressed the Reviewer’s concern regarding the logic of assessing sperm methylation, the cause-effect relationship in the *Mtrr^{gt}* model, and manuscript content in responses to Reviewer 2 (points [0.1] and [0.2]) and Reviewer 3 (point [0.1]). We agree that the text/data regarding Figs. 5 and 6 in the original manuscript was unclear in places and we have now better defined our rationale, data and associated conclusions (see lines 273-86, 288-346, 419-28 of revised manuscript).

We disagree that the HIRA data shows only ‘weak’ associations, as there are several independent lines of evidence to indicate the potential importance of HIRA in the *Mtrr^{gt}* model: i) the *Hira* DMR overlaps with other genetic and developmentally relevant epigenetic features to suggest its importance in transcriptional regulation (Fig. 5a), ii) DNA methylation in *Hira* sperm DMR is responsive to the severity of the *Mtrr^{gt}* genotype (Fig. 5c), iii) the pattern of *Hira* RNA dysregulation is similar to the pattern of maternal phenotype inheritance in the *Mtrr^{gt}* model as determined by assessing two separate genetic pedigrees (Fig. 6). Transcriptional memory has been described in another intergenerational model (Radford et al, 2014 *Science*) but we show that it associates with phenotype and extends to the F3 generation, iv) *Hira*^{-/-} embryos phenocopy embryos in the *Mtrr^{gt}* model (Roberts et al, 2002 *Mol Cell Biol*; Padmanabhan et al, 2013 *Cell*) suggesting a similar underlying mechanism, v) the

function of HIRA as a histone chaperone and in ribosome assembly fit well with potential TEI mechanisms involving nucleosome spacing and ribosome heterogeneity (Torre et al, 2020 *Nat Struct Mol Biol*; Lin et al, 2014 *Dev Cell*; Zhang et al, 2019 *Nat Rev Endocrinol*), and vi) HIRA function is important at the key junction between generations (germ cell and zygotes) as it is key for maintaining chromatin integrity in oocytes and later for remodelling the paternal pronucleus after fertilization (Nashun et al, 2015 *Mol Cell*; Pchelintsev et al, 2013 *Cell Rep*). Based on the Reviewer's comment, we have clarified the above points in lines 289-330, 425-8 of the revised manuscript.

Ultimately, we have shown the power of our MeDIP-seq analysis in identifying potential players in the TEI mechanism and perfectly set up the next line of questioning. Importantly, our data provides some clarity for why phenotypic inheritance occurs via the maternal lineage in the *Mtrr^{gt}* model: an ooplasm-inherited factor (like HIRA) might lead to greater phenotypic severity and epigenetic instability than a sperm-inherited factor (Fig. 7, lines 327-30, 429-34). See also response to Reviewer 3, point 7 for further comments about the data in Fig. 6.

2. Figure 2 – the sex of embryos is not listed. And the impact of litter effects is also not included. In many cases throughout the paper, the N's are quite small (N=3), so litter effects and clarity in embryo and placenta sex is essential.

Authors' response [2]: Fig. 2 showcases sperm rather than embryos. We think that the Reviewer was instead referring to Fig. 6. **Conceptus sex** was variable among groups in all figures where embryos and placentas were assessed. Where possible, we tried to assess equal numbers of males and females per experimental group. This was not always possible given the sheer volume of samples required in this study and it was often difficult to find sufficient numbers of severely affected conceptuses, which in some cases have little material (e.g., DNA, RNA) to work with. Importantly, we previously and extensively investigated sexual dimorphism of developmental phenotypes in the *Mtrr^{gt}* mouse line (Padmanabhan et al, 2017 *Envir Epigen*). We concluded that there are no sex-specific effects at E10.5 in *Mtrr^{gt/gt}* conceptuses or in F1-F4 generation *Mtrr^{+/+}* conceptuses derived from either an F0 *Mtrr^{+/gt}* male or female. Additionally, where possible, we have determined that there was no significant methylation or gene expression difference between sexes at E10.5. This is consistent with our previous analyses of several imprinted loci where no sex difference at E10.5 was observed (Padmanabhan et al, 2013 *Cell*). We have included in the revised manuscript that both males and female conceptuses were assessed and that there is no sexual dimorphism at E10.5 (lines 480-1).

An important feature of the *Mtrr^{gt}* mouse line is the wide spectrum and frequency of phenotypes that appear not only within one litter but also between litters (Padmanabhan et al, 2013 *Cell*). This might be characterised as a '**litter effect**', which most certainly exists and was discussed at length in Padmanabhan et al, (2013 *Cell*) and Padmanabhan et al, (2017 *Environ Epigenet*). We hypothesize that the inter- and intra-litter phenotypic variability is caused by stochastic changes in the germline epigenome leading to different phenotypes in each individual. We have highlighted this point in the revised manuscript (lines 382-5). Litter size did not associate with fetus or placenta

weight at E10.5 in any of the pedigrees (Padmanabhan et al, 2017 *Environ Epigenet*). The revised manuscript was updated accordingly (lines 497).

We disagree with the comment that “in many cases throughout the paper, the Ns were quite small (N=3)” because our **N numbers** in most cases were higher than this. We apologise that this information was unclear in the original manuscript. Nearly all instances where N=3 pertained to the ‘severely affected’ phenotypic group whereby the embryos/placentas were less frequent during dissections (Padmanabhan et al, 2013 *Cell*) and were generally smaller with less material (e.g., DNA, RNA). We have carefully confirmed all N values in all figure legends for each experiment and clarified what group specifically had N=3 (see lines 832-52 for Fig. 6 in particular). We found one typo in the legend of Fig. 5c, which stated “N=3 males/group”. Instead, N=4-8 males in this bisulfite pyrosequencing experiment and we have corrected the mistake on a genotype specific basis (see lines 826-9). Beyond this, we have converted all bars graphs throughout the manuscript to ‘scatter plot with bar’ graphs to more clearly show individual data points in each experiment and as a result, the N is clearer for each data set.

3. In many cases, the authors have listed effects or outcomes by ‘affected’ or ‘severely affected’, which could potentially bias the results. How are the authors controlling for the bias of inclusion of severely affected offspring, when these are more likely to be closer to death, and hence any gene expression or changes being examined would seem questionable for the health of the cells. Again, this brings up the concern for distinguishing between cause and effect.

Authors’ response [3]: We rigorously scored conceptuses and placed them into only one of several phenotypic categories for the purpose of better understanding the molecular pathways disrupted in each phenotypic group (see Methods for details, lines 474-95). This phenotyping regime was specifically designed to *avoid* bias since the data from phenotypically normal and severely affected conceptuses were never pooled. If phenotypic groups were not separated and, instead, molecularly assessed together, the result would not be meaningful and more likely to create bias.

Our phenotypic categories distinguished between severely affected conceptuses (those with one or more phenotype including failure of neural tube closure (cranial or spinal cord), pericardial edema, reversed heart looping, enlarged heart, twinning, hemorrhages, eccentric and/or lack of chorioallantoic attachment, etc.) and resorbed conceptuses (dead). The resorbed conceptuses were never molecularly assessed due to a lack of healthy fetally-derived cells. This point was clarified in the revised manuscript on lines 488-91. Furthermore, several of the severe phenotypes characterised appear later in development at E14.5 and E18.5 suggesting that these fetuses were alive at E10.5 (E. Watson, manuscript in preparation). A significant proportion of severely affected conceptuses had normal embryo and placental weights at E10.5, and demonstrated a beating heart (though, this is not always a reliable indicator based on the fact that dissections occurred in cold 1x PBS, which stops the heart beating). Even if the embryos were to ‘near death’, our molecular analysis provides an interesting perspective of the molecular pathways and mechanisms just prior to the demise of the conceptus, particularly when compared to phenotypically normal conceptuses. The field of developmental biology is based on disrupting genetic

pathways to determine the molecular and phenotypic effects. We do not think that our experimental design differs from what is a standard line of questioning in the field and in fact, directly investigates 'cause and effect'. Again, we kindly refer the Reviewer to our responses to Reviewer 2, point 0.1 and Reviewer 3, point 0.1.

4. Alters differential placental DNA methylation – but what does this mean?

Authors' response [4]: We cannot find the specific line in the text where this phrase was used and perhaps the Reviewer was instead referring to line 72 (“...alters differential DNA methylation in the germline.”). We have changed this phrase to “...we show that germline DNA methylation is altered in F0 *Mtrr*^{+/*gt*} males.” on line 76-7 of the revised manuscript.

5. The authors are largely focused on examining transgenerational epigenetic inheritance –exclusively on DNA methylation. But what is missing is evidence linking this inheritance to any phenotype – how are these related in this paper? The authors need to distinguish between these two.

Authors' response [5]: Ultimately, this Reviewer's comment is a continuation of their concern of a lack of 'cause and effect', and we have addressed this issue at length above (see responses to Reviewer 2, point 0.1 and Reviewer 3, point 0.1). We reiterate that we previously published the causal relationship between an *Mtrr*^{+/*gt*} mouse (male or female) on the phenotypic outcome in the subsequent F1-F4 generations and demonstrate that it is a true model of TEI (see Padmanabhan et al, 2013 *Cell*).

The goal of this manuscript (in addition to other projects in our lab, and certainly the TEI field at large in their respective models) is to elucidate the epigenetic mechanism behind the TEI phenomenon. Currently, very little is known and unfortunately, it is a slow moving field given the nature of the phenomenon. A specific *epigenetic* link to mammalian transgenerational *phenotypic* inheritance is yet-to-be found, and so this is a high standard to hold our manuscript to. We are very excited that our manuscript provides epigenetic evidence that strengthens the *Mtrr*^{*gt*} model of TEI by showing germ cell epigenetic instability, transcriptional memory of sperm DMRs that persists generationally further than others have demonstrate and that is associated with phenotypic inheritance, and a potential factor (i.e., HIRA) was hypothesized as 'a mediator' of TEI for further analysis in future manuscripts. Therefore, our study is important because we show how the *Mtrr*^{*gt*} model can be used to study TEI mechanisms and we have opened up new avenues of research.

We chose to focus on DNA methylation among the other epigenetic candidates of TEI (e.g., histone modifications, sncRNA content) for the following reasons: i) one-carbon metabolism is directly linked to cellular methylation including DNA methylation, ii) our previous observations that showed DNA methylation changes associated with gene misexpression in F2 *Mtrr*^{+/*+*} placentas, an effect that becomes more severe with severity of phenotype (Padmanabhan et al, 2013 *Cell*), and iii) the technical expertise of the lab. Importantly, our current manuscript uses genome-wide DNA methylation patterns to indicate that germ cells of the F0 generation of the *Mtrr*^{*gt*} model are

epigenetically unstable and that some of these disrupted regions (e.g., *Hira*, *Cwc27*, *Tshz3*) are associated with transcriptional memory in somatic cells of subsequent wildtype generations in a manner that aligns with maternal phenotypic inheritance. Even though DMRs are reprogrammed in offspring somatic tissue, some DMRs are inherited/reconstructed in offspring *germ cells* since common DMRs were observed in sperm from *Mtrr*^{+/+} males and the *Mtrr*^{+/*gt*} fathers (see lines 385-8). The functional relevance of this remains unclear. However, it is clear that other epigenetic mechanisms are likely involved in TEI in concert with DNA methylation, and our DNA methylation data highlight potential regions to focus on in future studies (see lines 409-38). Using our DNA methylation data, we also identified HIRA, a potential epigenetic regulator that is perfectly situated to play a role in TEI. This will also be addressed in future analyses. Overall, the exploration of DNA methylation as a potential epigenetic regulator of TEI is justifiable in a proven model of TEI.

6. Figure 6 western blots - the authors need to show the bands.

Authors' response [6]: As requested, we have moved the western blots from Supplementary fig. 11 of the original manuscript into Fig. 6 of the revised manuscript. The figure legends and citations in the text have been changed accordingly. See also response to Reviewer 2 (point 3.2) for information about how the western blot data was quantified.

7. Figure 6 data overall are quite perplexing and seemingly weak evidence to support any connection between the intergenerational effects with a lncRNA and *Hira*. The data do not show this pattern in many cases, and it is a bit of a stretch to infer causality. Further, there is no demonstration of any functional outcome for potential changes in *Hira*. Where is any histone functional change?

Authors' response [7]: We have addressed the Reviewer's concern regarding the 'seemingly weak' association between our *Hira* data and phenotypic inheritance in our response to Reviewer 3, point 1. We acknowledge that the presentation of this data was unclear in the original manuscript and we have edited the text and Fig. 6 accordingly (please see lines 288-346 and Fig. 6 of the revised manuscript). It is important to note that our data shows a *transgenerational* rather than *intergenerational* association (as indicated by the Reviewer) of *Hira* RNA expression and phenotypic inheritance based on the fact that we see altered *Hira* expression in the F2 and F3 wildtype embryos derived from an F0 *Mtrr*^{+/*gt*} male and the F3 wildtype embryos derived from an F0 *Mtrr*^{+/*gt*} female (see our response to Reviewer 3, point 8 below for our definition of TEI). Therefore, the extent of transcriptional memory is one generation further than previously reported in other models (Radford et al, 2014 *Science*).

We argue that our data shows a functional outcome since we assessed two separate genetic pedigrees that indicate a similar correlation between the patterns of *Hira* RNA misexpression and maternal phenotypic inheritance. While important, we think that it is outside the scope of this manuscript to explore the role of *Hira* in TEI. This will require a fairly substantial (and expensive) experimental undertaking including an H3.3 ChIP-seq experiment, ideally on germ cells or zygotes, to understand the extent to

which histone incorporation is altered by the *Mtrr^{gt}* mutation over multiple generations. We have preliminary immunofluorescence data of C57Bl/6J and *Mtrr^{gt/gt}* zygotes shows a similar intensity of H3.3 immunostaining in the maternal and paternal pronuclei of control and *Mtrr^{gt/gt}* zygotes (C. Handford, G. Nishimura, E. Watson, unpublished). However, the mutant zygotes showed a higher degree of inter-individual variability. Therefore, it will be important to assess genome-specific localization of H3.3 incorporation using single cell sequencing technology. Additionally, there are a number of associated experiments that put this entire line of questioning outside of the scope of this manuscript. As Reviewer 3 noted in point 1, the manuscript is quite data heavy as it is. Indeed, the next 'chapter' of this exciting story will focus on exploring the functional role of HIRA in TEI. We have now indicated this as a future area of research in the revised manuscript (lines 425-428).

8. Is it really TEI if each subsequent generation requires an exposure to prior generation change in folate metabolism resulting from their development and intrauterine environment? Does the current data rule that out? This should be discussed.

Authors' response [8]: The Reviewer asks whether the *Mtrr^{gt}* mouse line is a true model of TEI. The answer is yes and this has been extensively assessed in Padmanabhan et al, (2013 *Cell*), addressed in our responses above (Reviewer 2, points 0.1 and 0.2, and Reviewer 3, points 0.1, 5, and 7), and further clarified in this response. Most importantly, the current study does not rule out TEI of phenotypes in the *Mtrr^{gt}* model but rather explores the mechanism(s) behind it.

The Reviewer asks the extent of 'exposure' to abnormal folate metabolism that occurs in the *Mtrr^{gt}* model and whether the uterine environment plays a role given that it is a model of maternal grandparental TEI. These are good questions. First, to ensure that we share the same **definition of TEI**, we have defined it here (see also Blake and Watson, 2016 *Curr Opin Chem Biol*). Generally, mammalian TEI occurs after the F0 generation is exposed to an environmental stressor leading to phenotypes in subsequent generations in the absence of the stressor. The mechanism is unknown. The Reviewer is correct: continued exposure of a stressor over multiple generations does not constitute TEI. However, there are sex-specific complexities when defining stressor exposure and the initiation of TEI. In exposed F0 males, the F1 generation is exposed in the testis and/or reproductive tract as a F0 sperm. Therefore, it is not considered TEI via the germline until a phenotype arises in the F2 generation (etc.). In exposed F0 females, the F1 generation is exposed as an F0 oocyte and potentially as a fetus in the F0 uterus. Furthermore, the F2 generation is potentially exposed as a F1 fetal germ cell in the F0 uterus. Therefore, in this case, it is not considered TEI via the germline until phenotypes arise in the F3 generation. We have clarified this definition in the context of the *Mtrr^{gt}* model on lines 29-41 of the revised manuscript.

In Padmanabhan et al, (2013 *Cell*), we showed that the *Mtrr^{gt}* model displays TEI, by performing highly controlled genetic pedigrees so that **only the F0 generation carried the *Mtrr^{gt}* allele**. All subsequent generations were genetically wildtype (see schematics of genetic pedigrees in Supplementary fig. 1b). We have also **ruled out inherited disruption of one-carbon metabolism** (via mass spectrometry) since

plasma homocysteine concentrations were normal in F1 and F2 wildtype mice (Padmanabhan et al, 2018 *J Phys*; E. Watson, unpublished data). We showed that the *Mtrr^{+gt}* genotype in the F0 generation causes a wide spectrum and frequency of developmental phenotypes in the F2, F3, and F4 wildtype generations regardless of whether the F0 *Mtrr^{+gt}* mouse was a male or female (Padmanabhan et al, 2013 *Cell*). **The presence of phenotypes in the F3 and F4 wildtype generations supports TEI, according to the definition above.**

Since it is a model of *maternal* grandparental TEI, we explored the effects of the F1 wildtype uterine environment on phenotypic inheritance as an alternative mechanism to TEI by performing an **embryo transfer experiment** (Padmanabhan et al, 2013 *Cell*). In this case, F2 wildtype blastocysts derived from either *Mtrr^{+gt}* maternal grandparent were flushed out of the oviducts/uteri of the F1 wildtype female and transferred into uteri of control females. The result was complex because some of the phenotypes (e.g., growth defects) were rescued and other phenotypes (e.g., congenital malformations) persisted suggesting two separate epigenetic mechanisms. In other words, ***Mtrr^{gt}* allele in mice resulted in epigenetic effects both on the wildtype daughter's uterine environment (not TEI: growth phenotypes) and on her gametes (TEI: congenital malformations) leading to defective fetal and placental development in the wildtype grandprogeny** (Padmanabhan et al, 2013 *Cell*). See also lines 39-41 in the revised manuscript. Correspondingly, the only phenotypes present in the F3 and F4 generations of non-transfer *Mtrr^{+gt}* maternal grandparental pedigrees were those caused by germ cell inheritance (i.e., congenital malformations).

While it would be ideal to reiterate all of this information in the current study, there simply is not room. We have provided background information where necessary and otherwise cited the original studies. We have edited the introduction of the revised manuscript to clarify key points about the model (e.g., lines 37-41, 59-73).

9. Focus on Hira following the sperm methylation studies is awkward. Authors could develop this better. Fig 3 shows no change in DMR for Hira? Doesn't show up changed until Fig 4 for expression in the F2 embryo.

Authors' response [9]: We agree that this section of the manuscript was difficult to follow and we have now edited it in the revised manuscript to clarify how the data is linked (see lines 288-317). Also, please refer to our response to Reviewer 3, points 1 and 7 for further discussion on the data in Fig. 6.

10. The introduction needs to explain the *Mtrr-gt* more clearly – as a reviewer, I had to search back in the literature to figure out what it was. As it is the major focus of the paper, reader shouldn't have to search for what it is or how it was made and validated. The authors could make a more clear explanation as to what the change in folate metabolism and what parent came from – maybe in a schematic? Getting lost in the weeds of genotype, but not clear what the authors hypothesis is related to exposure for germ cell and reproductive pathway is.

Authors' response: We are sorry for the Reviewer's frustration and we thank them for their honesty. We are conscious of the fact that many of Reviewer 3's comments were about our published data rather than the current data. We appreciate that the model is complex and that there is a lot of background information to flag. A major challenge in writing this manuscript was to stay within the word count mandated by the journal. Based on the Reviewer's concern, we now have carefully gone through the introduction (and the whole manuscript) to clarify key points about the model (e.g., lines 29-84). We would like to bring to the Reviewer's attention Supplementary Fig. 1 (previously Supplementary Figs. 1 and 2), which shows schematics of folate metabolism and the breeding schemes used in this manuscript and in previous published work. Also, we previously included in the Methods information about how the mutation was generated (see lines 450-59 of revised manuscript) and the breeding scheme (see lines 464-71 of revised manuscript).

11. The authors have previously shown that *Mtrr* +/- *gt* male mice have reduced homocysteine, not hyper-, how does that fit into this story? These males' sperm are being affected by their genotype and folate metabolism in seemingly very different, opposing ways.

Authors' response: In Padmanabhan et al (2013 *Cell*, Fig 1D), we showed that there was a statistically significant difference in plasma total homocysteine concentrations between C57Bl/6J males ($5.7 \pm 0.6 \mu\text{M}$; mean \pm sd) and *Mtrr*^{+/*gt*} males ($4.3 \pm 0.6 \mu\text{M}$; $p=0.047$ as determined by an independent t test). This experiment was not ideal because $N=3$ for each experimental group, and an incorrect statistical analysis was performed. We re-analysed the data by a Kruskal-Wallis with a Dunn's multiple comparisons test, which indicated that there was no significant difference between C57Bl/6J and *Mtrr*^{+/*gt*} males (see below in Fig. R3.1).

Regardless, the Reviewer asks why a single *Mtrr*^{gt} allele is sufficient cause TEI when they are metabolically 'normal'. The short answer is that we do not know. We previously showed that spermatogenesis and sperm function were normal in *Mtrr*^{+/*gt*} males compared to control C57Bl/6J males (Blake et al, 2019 *Reprod Fertil Dev*; refer to our responses to Reviewer 3, points 12 and 13) and that there was no direct paramutation effect of the *Mtrr*^{gt} allele (Figure S4 of Padmanabhan et al, 2013 *Cell*) similar to other models (Rassoulzadegan et al, 2006 *Nature*). It is possible that because *Mtrr*^{+/*gt*} mice derive from *Mtrr*^{+/*gt*} intercrosses, they exhibit a degree of epigenetic instability inherited from their parents. Alternatively, *Mtrr*^{+/*gt*} mice might be metabolically abnormal and measuring plasma homocysteine alone is insufficient to understand the extent metabolic disruption. We have added a sentence to the discussion of the revised manuscript to address this point (see lines 396-8).

Fig. R3.1. Plasma total homocysteine concentrations in C57Bl/6J (black), *Mtrr*^{+/*+*} (purple), *Mtrr*^{+/*-*} (green) and *Mtrr*^{gt} adult male mice as determined by underivatized LC-MS/MS. $N=3$ males per experimental group. Data is presented as mean \pm sd. * $p<0.05$ Kruskal-Wallis test followed by a Dunn's multiple comparison test. (Reanalyzed data from Padmanabhan et al, 2013 *Cell*)

12. For sperm assay methods, it appears no swim-up collection or assessment for motility was done. How are the authors confirming live cells or changes in sperm functionality? Has this already been examined – using IVF or ICSI?

Authors' response: We used a 'swim-up' method during the sperm collection process. In our methods section of our original manuscript, we cited a protocol (Hisaro et al, 2013 *Nat Protocols*) rather than stating this fact outright. We have now clarified the method that was used in the revised manuscript (see line 501).

We have extensively analysed spermatogenesis and sperm functionality in the *Mtrr^{gt}* model, which we published in Blake *et al* (2019 *Reprod Fertil Dev*). In summary, we showed that spermatogenesis occurred in a normal manner in *Mtrr^{+/+}*, *Mtrr^{+/gt}* and *Mtrr^{gt/gt}* testes. Mature spermatozoa from each male genotype showed normal morphology, motility, viability and function and all males assessed showed normal fertility. We have more clearly indicated this point in the revised manuscript (see line 126; also refer to our response to Reviewer 3, point 13). This data reassures us that the defect inherited from sperm from *Mtrr^{+/gt}* males is epigenetic in nature (e.g., chromatin marks or RNA content) rather than developmental. We do not think that performing ICSI or IVF will alter our conclusions about sperm development or functionality, particularly when we do not observe a defect.

13. Determination of 'abnormal' for embryo phenotype – if reproductive processes/implantation etc. are altered by genotype, does that change interpretation of outcomes for role of Mtrr? Authors should discuss fertility outcomes – is there an effect on rate of implantation or fertilization? Sex ratio of offspring with phenotype or abnormal outcome?

Authors' response: It is likely that fertilization and blastocyst implantation occurs at a normal frequency in the *Mtrr* maternal grandparental pedigrees and in *Mtrr^{gt/gt}* intercrosses. This is because normal litter sizes are observed at midgestation (E10.5) compared to C57Bl/6J controls (Padmanabhan et al, 2013 *Cell*), which indicates that most ovulations are fertilized, develop to blastocyst stage and implant into the uterus to appear at E10.5. Preliminary analysis of blastocyst structure and lineage segregation in *Mtrr^{gt/gt}* embryos also indicates that this process occurs as normal (N. Capatina, H.W. Yung, and E. Watson, unpublished data). We are currently writing a manuscript that assesses the effects of the *Mtrr^{gt}* allele on development prior to E10.5 (A. Wilkinson, E. Watson, in preparation), and think that this discussion is outside of the scope of the current manuscript. We previously published that *Mtrr^{+/gt}* and *Mtrr^{gt/gt}* males display normal fertility rates, that their spermatozoa develop/mature as expected, and their sperm are normal in structure and motility (Blake et al, 2019 *Reprod Fertil Dev*) (see line 126; refer to our response to Reviewer 3, points 11-12). There is evidence that *Mtrr^{+/+}*, *Mtrr^{+/gt}* and *Mtrr^{gt/gt}* females might take longer to establish a copulatory plug (uninterested in mating?), but those that do mate are most often pregnant at E10.5 (Blake et al, 2019 *Reprod Fertil Dev*), which indicates normal fertility. In the end, our robust phenotypic readout at E10.5 is sufficient to assess a multigenerational effects of the *Mtrr^{gt}* allele. We have previously published that there is no sexual dimorphism of

phenotypes observed at E10.5 in any of the *Mtrr* pedigrees (Padmanabhan et al, 2017 *Environ Epigen*; see also our response to Reviewer 3, point 2 (paragraph 2)). None of these data alters the fact that the *Mtrr^{gt}* model is a true model of TEI (see our response to Reviewer 3, point 8 and Padmanabhan et al, 2013 *Cell*).

14. Why so much variance of penetrance here? Could there be an interactive effect of intrauterine position? Was intrauterine position controlled for in placenta or embryo studies here?

Authors' response: We hypothesize that the **variance of phenotypic penetrance** observed in the *Mtrr^{gt}* mouse line at E10.5 is caused by stochastic epigenetic changes in each individual whereby the regulation of developmentally important genes is affected differently in each individual leading to a wide range of phenotypes (see also lines 382-4 of the revised manuscript). The data in this manuscript and in our previously published work (Padmanabhan et al, 2013 *Cell*) demonstrates potential 'hot spots' for epigenetic change (i.e., the common DMRs between individuals of a common genotype and between different genotypes) but also there are other regions that are variably methylated between individuals (T. Bertozzi, E. Watson, A. Ferguson-Smith, manuscript submitted) that might lead to more subtle effects causing intra-individual phenotypic differences.

It is unlikely that **uterine position** of the implantation site (i.e., close to ovary versus close to cervix) relates to phenotypic risk in the *Mtrr^{gt}* model. A lack of phenotypes in C57Bl/6J controls (Padmanabhan et al, 2013 *Cell*) suggests that uterine position does not increase risk for congenital abnormalities at E10.5 in normal pregnancy. In *Mtrr^{gt}* mouse line, there were litters in which every implantation site fit into a phenotypic category other than normal, and some litters where the entire litter was phenotypically normal regardless of uterine position. We do not have the data available to do this analysis, and we think that this is outside the scope of this manuscript. We previously **controlled for litter size** and no correlation with phenotypic risk was observed (Padmanabhan et al, 2017 *Environ Epigen*) (see line 497).

It is clear that the uterine environment of F1 *Mtrr^{+/+}* females (derived from an F0 *Mtrr^{+/gt}* male or female) plays a role in the establishment of growth phenotypes (e.g., growth restriction, growth enhancement, developmental delay) because these phenotypes were specifically rescued by an embryo transfer experiment (Padmanabhan et al, 2013 *Cell*). The exact mechanism is unclear though it is likely epigenetic since it is caused by *Mtrr^{gt}* heterozygosity in either one of her parents. Gross morphological and molecular analyses suggest that uterine structure and decidualization occurs normally in females from the *Mtrr^{gt}* pedigrees (A. Wilkinson, K. Menelaou, E. Watson, in preparation).

15. Seems to be a large literature of paternal studies missed in the authors' citations for many groups focusing on sperm RNAs and intergenerational inheritance.

Authors' response: As suggested, we now have added more information about alternative mechanisms involved in direct epigenetic inheritance via the paternal lineage (see lines 391-6, 409-418 in the revised manuscript).

REVIEWER COMMENTS

Reviewer #1 (Remarks to the Author):

I'm satisfied with the revision and support publication

Reviewer #2 (Remarks to the Author):

The authors have answered most of my comments satisfactorily and their revised manuscript is improved in my opinion. I would suggest the authors consider a few changes to the manuscript before publication.

Related to my main concern of focusing the study on sperm from F0. I admit being confused by the complexity of the breeding scheme and phenotypic distribution. If my understanding is correct the effect is not transmitted through the male lineage? If so I would state it somewhere in the description of the model. Also for a non-specialist reader not familiar with breeding scheme it would be easier if the suppl fig 1 legend would indicate that circle are female and square are male...

I still think that the fact that no phenotype (or a mild phenotype as described in their previous work, now mentioned in the present manuscript) is apparent in F1 from mutant male father suggests that the mechanism leading to TEI initiating might rely on different epigenetic cues than the one perpetuating the effect through the female lineage. The authors do touch on this issue in the manuscript and that should help the reader.

Important other points:

I found a number of inaccurate statements:

(i) Related to Hira role in the transcription of rRNA the authors refer in several instances to a role for Hira in ribosomal assembly (l84 p4; l294 p12;l425p16). I only briefly checked ref 27 but it seems to be that Hira is only involved in the production of a ribosomal component (the rRNA) rather than assembly/function of ribosomes per se.

(ii) l 282 p11, H3K4me3 is not known to be a repressive mark. So should replace "one or more repressive histone marks" by e.g. methylated histones

(iii) I really don't think that the authors can substantiate claim in introduction "exhibit transcriptional memory of germline methylation defects" l 24 p2, results "Transcriptional memory of sperm DMRs" l293 p11, and discussion "This transcriptional memory of a germline DMR persisted" l360 p14. These statements really give the impression that the transcriptional effects are caused by the observed DNA methylation changes when the study shows association/correlation but does not provide a test for causality. I strongly suggest a change of wording that would reflect this.

Reviewer comments:**Reviewer #1 (Remarks to the Author):**

I'm satisfied with the revision and support publication

Reviewer #2 (Remarks to the Author):

The authors have answered most of my comments satisfactorily and their revised manuscript is improved in my opinion. I would suggest the authors consider a few changes to the manuscript before publication.

Reviewer comment [2.1]: Related to my main concern of focusing the study on sperm from F0. I admit being confused by the complexity of the breeding scheme and phenotypic distribution. If my understanding is correct the effect is not transmitted through the male lineage? If so I would state it somewhere in the description of the model. Also for a non-specialist reader not familiar with breeding scheme it would be easier if the suppl fig 1 legend would indicate that circle are female and square are male... I still think that the fact that no phenotype (or a mild phenotype as described in their previous work, now mentioned in the present manuscript) is apparent in F1 from mutant male father suggests that the mechanism leading to TEI initiating might rely on different epigenetic cues than the one perpetuating the effect though the female lineage. The authors do touch on this issue in the manuscript and that should help the reader.

Authors' response [2.1]: As requested by the editor and Reviewer 2, we have clarified information about the breeding scheme and phenotypic distribution in the main text of the revised manuscript (including lines 51-93 (specific text highlighted), 263-267, 347-354, 365-6) and more comprehensively in an updated Supplementary fig. 1 (as indicated above), which now includes the sex in the pedigree legend. Overall, we think that these changes improve the manuscript.

Furthermore, we clarified the text to indicate that the phenotypes at E10.5 are not transmitted through the male (lines 72, 74-76), and further emphasise that the initiating mechanism of TEI might rely on different epigenetic cues in the maternal grandmother versus the maternal grandfather (lines 91-93). We are pleased that the Reviewer thinks that the added discussion on this latter point is now sufficient.

Reviewer's comment [2.2]: Important other points: I found a number of inaccurate statements:

(i) Related to Hira role in the transcription of rRNA the authors refer in several instances to a role for Hira in ribosomal assembly (l84 p4; l294 p12;l425p16). I only briefly checked ref 27 but it seems to be that Hira is only involved in the production of a ribosomal component (the

rRNA) rather than assembly/function of ribosomes per se.

Authors' response [2.2.i]: We thank the reviewer for this careful reading. These statements have now been clarified in the revised manuscript (lines 104-5, 321, 460).

(ii) l 282 p11, H3K4me3 is not known to be a repressive mark. So should replace "one or more repressive histone marks" by e.g. methylated histones

Authors' response [2.2.ii]: We have changed the text as requested (line 308).

(iii) I really don't think that the authors can substantiate claim in introduction "exhibit transcriptional memory of germline methylation defects" l 24 p2, results "Transcriptional memory of sperm DMRs" l293 p11, and discussion "This transcriptional memory of a germline DMR persisted" l360 p14. These statements really give the impression that the transcriptional effects are caused by the observed DNA methylation changes when the study shows association/correlation but does not provide a test for causality. I strongly suggest a change of wording that would reflect this.

Authors' response [2.2.iii]: We thank the reviewer for this comment and we have now toned down all conclusions in the revised manuscript, particularly those flagged by the reviewer (lines 24, 102, 289, 292, 360, 391-2, 453-4, 473).